

# A note on the identity module in $c = 0$ CFTs

Yifei He[1,2] and Hubert Saleur[2,3]

**1** Institut Philippe Meyer, École Normale Supérieure, Université PSL,
24 rue Lhomond, F-75231 Paris, France
**2** Université Paris-Saclay, CNRS, CEA, Institut de Physique Théorique,
91191, Gif-sur-Yvette, France
**3** Department of Physics, University of Southern California,
Los Angeles, CA 90089, USA

## Abstract

It has long been understood that non-trivial Conformal Field Theories (CFTs) with vanishing central charge ($c = 0$) are logarithmic. So far however, the structure of the identity module – the (left and right) Virasoro descendants of the identity field – had not been elucidated beyond the stress-energy tensor $T$ and its logarithmic partner $t$ (the solution of the "$c \to 0$ catastrophe"). In this paper, we determine this structure together with the associated OPE of primary fields up to level $h = \bar{h} = 2$ for polymers and percolation CFTs. This is done by taking the $c \to 0$ limit of $O(n)$ and Potts models and combining recent results from the bootstrap with arguments based on conformal invariance and self-duality. We find that the structure contains a rank-3 Jordan cell involving the field $T\bar{T}$, and is identical for polymers and percolation. It is characterized in part by the common value of a non-chiral logarithmic coupling $a_0 = -\frac{25}{48}$.

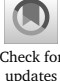

## 1  Introduction

While conformal field theory (CFT) in 2 dimensions has garnered an enviable list of successes, several of its most exciting potential applications are not yet under full control. This is the case in particular of the description of critical points in disordered systems (both in 2 and 2+1 dimensions), and of geometrical critical problems such as polymers (self-avoiding walks) and percolation. The difficulty in tackling these problems originates from the lack of unitarity, itself inherited from disorder averaging or non-local constraints such as self-avoidance of random walks. As it turns out, non-unitarity can have daunting consequences, the full extent of which we are just starting to appreciate.

Geometrical critical problems encompass the large class of loop and cluster models, for which quite a bit of progress has been obtained recently, in part due to systematic use of bootstrap techniques. It is now clearly understood that the corresponding theories are generically logarithmic [1–4], and exhibit some kind of "interchiral symmetry" [5], with some truly degenerate (in the Virasoro sense) fields, and spectra of critical exponents with Kac labels including most rationals [6].

The specific cases of polymers and percolation are however more complicated than the generic loop or cluster model because they occur right at central charge $c = 0$ (a feature shared by most disorder related problems). There - like at any other "rational point" - the logarithmic(L) CFTs [7] describing generically polymers or percolation develop a considerably more intricate logarithmic structure, with Jordan blocks of arbitrary rank, and indecomposable modules with highly complex Virasoro structures. The bootstrap approach to correlation functions meanwhile encounters many divergences - a natural manifestation of the representation theoretic "mixing" of Virasoro modules.

Nonetheless the case $c = 0$ is fascinating for physical reasons, and we know more about it than for the other rational points. A very insightful first attempt to understand theories with this value of the central charge dates back to [8,9] with the introduction of $t$ – the "logarithmic partner" of the stress energy-tensor, together with its "logarithmic coupling" $b$. It took another decade for this coupling to be observed numerically in [10,11], hence providing some concrete insight into what had been so far a very abstract notion. More in depth study of lattice models showed that the predictions in [9] were slightly off, and that the $b$-numbers for percolation and polymers were in fact $b_{perco} = -\frac{5}{8}$ and $b_{poly} = \frac{5}{6}$ respectively. Further work also brought to attention the profound difference between the structures proposed in [9] - which apply really to the boundary case - and those for the bulk theories. For the latter, it was found that $b = -5$ *both* for percolation and polymers [12].

While the full bulk theory for these problems has attracted special attention - and some progress has definitely been made [13], [14], [15], even the simplest question of how to properly complete the ideas proposed in [9] in the full non-chiral case has remained open until now. The corresponding structure of the "identity module" - that is how $t, T, \bar{t}, \bar{T}, T\bar{T}, t\bar{T}, T\bar{t}, \ldots$ are

related with each other under the action of the left and right Virasoro algebras (we will in general denote the enveloping algebra of the product $\text{Vir} \otimes \overline{\text{Vir}}$ by $\mathcal{V}$), how they contribute to a general OPE etc - is the subject of this paper.

We note that earlier attempts have been made to tackle this problem - see in particular the conjecture in [15]. The novel ingredient that allows us to make further progress here is the recent understanding of some four-point functions and OPEs for generic loop and cluster models [1,5,16,17].

The paper is organized as follows. In section 2, we revisit a time honored strategy [11, 18–20] by considering the behavior as $c \to 0$ of the generic OPE of two primary fields in the loop or cluster model, using the information recently obtained both about the spectrum and the existence of rank-two Jordan blocks for $L_0, \bar{L}_0$ [1,3,4] at generic values of $c$. By requiring the finiteness of two-point functions at $c = 0$, we obtain singularity cancellation conditions (2.13), (2.28) and (2.30), which allow us to establish the existence of a rank-three Jordan block of fields of weight $h = \bar{h} = 2$ when $c = 0$ with the bottom field being $T\bar{T}$, and determine the corresponding universal logarithmic coupling. Remarkably, this coupling turns out to be the same both for the percolation and polymers. In section 3, we consider the structure of the corresponding identity module under the action of the enveloping algebra of the left and right Virasoro algebras $\mathcal{V}$. This is done using general conformal invariance arguments and arguments of self-duality, leading to our main result in figure 2. Intriguingly, the structure we uncover is formally equivalent to the one proposed (for $c$ generic) in [1]. Various remarks are proposed in the conclusion while technical details (in particular the $c \to 0$ limit for the correlation functions of the order operator in the Potts model) are discussed in the appendix.

For the reader mostly interested in results, we give here the generic OPE of two diagonal primary fields at $c = 0$

$$
\begin{aligned}
\Phi_\Delta(z,\bar{z})\Phi_\Delta(0,\bar{0}) = (z\bar{z})^{-2\Delta}\Big\{ &1 + z^2\frac{\Delta}{b}\Big(t(0,\bar{0}) + T(0)\ln(z\bar{z})\Big) + \bar{z}^2\frac{\Delta}{b}\Big(\bar{t}(0,\bar{0}) + \bar{T}(\bar{0})\ln(z\bar{z})\Big) \\
&+ z\bar{z}^2\frac{\Delta}{2b}\partial\bar{t}(0,\bar{0}) + z^2\bar{z}\frac{\Delta}{2b}\bar{\partial}t(0,\bar{0}) + (z\bar{z})^2\frac{\Delta}{4b}\Big(\partial^2\bar{t}(0,\bar{0}) + \bar{\partial}^2 t(0,\bar{0})\Big) \\
&+ (z\bar{z})^2\frac{\Delta^2}{a_0}\Big(\Psi_2(0,\bar{0}) + \ln(z\bar{z})\Psi_1(0,\bar{0}) + \frac{1}{2}\ln^2(z\bar{z})\Psi_0(0,\bar{0})\Big) + \dots\Big\},
\end{aligned}
$$

where $(t, T)$ belong to a rank-two Jordan block:

$$
\begin{aligned}
\langle t(z,\bar{z})t(0,\bar{0})\rangle &= \frac{-2b\ln(z\bar{z}) + \theta}{z^4}, \\
\langle t(z,\bar{z})T(0)\rangle &= \frac{b}{z^4}, \\
\langle T(z)T(0)\rangle &= 0,
\end{aligned}
$$

and the fields $\Psi_i, i = 0, 1, 2$ belong to a rank-three Jordan block for $L_0, \bar{L}_0$:

$$
\begin{aligned}
\langle \Psi_2(z,\bar{z})\Psi_2(0,\bar{0})\rangle &= \frac{a_2 - 2a_1\ln(z\bar{z}) + 2a_0\ln^2(z\bar{z})}{(z\bar{z})^4}, \\
\langle \Psi_2(z,\bar{z})\Psi_1(0,\bar{0})\rangle &= \frac{a_1 - 2a_0\ln(z\bar{z})}{(z\bar{z})^4}, \\
\langle \Psi_1(z,\bar{z})\Psi_1(0,\bar{0})\rangle &= \frac{a_0}{(z\bar{z})^4}, \\
\langle \Psi_2(z,\bar{z})\Psi_0(0,\bar{0})\rangle &= \frac{a_0}{(z\bar{z})^4}, \\
\langle \Psi_1(z,\bar{z})\Psi_0(0,\bar{0})\rangle &= 0, \\
\langle \Psi_0(z,\bar{z})\Psi_0(0,\bar{0})\rangle &= 0,
\end{aligned}
$$

with the "logarithmic couplings"

$$b = -5, \;\; a_0 = -\frac{25}{48}.$$

Finally, we note that we have tried to strike a compromise between indicating the $c$-dependency of *all* the objects (like the stress-energy tensor $T$ or the field $X$, see below) we were considering, and keeping our equations somewhat readable. We believe the context should remove all possible ambiguities.

## 2 OPE and two-point functions

### 2.1 $c \to 0$ limit of Potts and $O(n)$ models

In contrast with what happens for rational minimal models (such as the Ising model), the action of the left and right Virasoro algebras in non-trivial $c = 0$ theories leads to modules which are indecomposable but not fully reducible - they do contain irreducible submodules, but cannot be written as direct sums of such modules. We will focus in this paper on the identity module - the module which contains the identity field $\mathbb{I}$ (with conformal weights $h = \bar{h} = 0$) "at the top" (i.e., $\mathbb{I}$ is the field with lowest conformal weight in the module). We recall that, both for the Potts and $O(n)$ models, the identity field is in fact the field with lowest conformal weight in the full theory. Moreover, it is the only field with $h = \bar{h} = 0$ - in contrast with, e.g. the theory proposed in [13].

The technical origin of these complicated features at $c = 0$ is that a descendant of the identity field – the stress-energy tensor $T(z), \bar{T}(\bar{z})$ – becomes null as $c \to 0$. More precisely, its Virasoro invariant norm-square (defined by using the standard conjugacy rule $L_n^\dagger = L_{-n}$) vanishes as $c \to 0$. It is however not possible to set $T = \bar{T} = 0$ in a physical theory (this would lead to the minimal model at $c = 0$, a trivial theory with only the identity field), and this implies, by a variety of arguments (see below), the appearance of a logarithmic partner of the stress-energy tensor, the famous $t$ field [9]. The Jordan blocks for $L_0, \bar{L}_0$ involving $t, \bar{t}, T, \bar{T}$ are now well understood for percolation and polymers, i.e., Potts model at $Q = 1$ and $O(n)$ model at $n = 0$ – both theories with $c = 0$. In order to see what happens for fields whose conformal weights $h, \bar{h}$ are both non-zero, we start from the Potts and $O(n)$ spectrum and Virasoro structure at generic $c$.

We focus on the conformal fields which will appear in the logarithmic mixing up to dimension $(h, \bar{h}) = (2, 2)$ in the limit $c \to 0$. This involves the following fields in the Potts and $O(n)$ spectra at generic $c$ (we also indicate their conformal dimensions at $c = 0$):

| Potts | $c \to 0$ | $O(n)$ | $c \to 0$ |
|---|---|---|---|
| $\mathbb{I} : (h_{1,1}, h_{1,1})$ | $\to (0,0)$ | $\mathbb{I} : (h_{1,1}, h_{1,1})$ | $\to (0,0)$ |
| $\bar{X} : (h_{1,2}, h_{1,-2})$ | $\to (0,2)$ | $\bar{X} : (h_{1,2}, h_{1,-2})$ | $\to (0,2)$ |
| $X : (h_{1,-2}, h_{1,2})$ | $\to (2,0)$ | $X : (h_{1,-2}, h_{1,2})$ | $\to (2,0)$ |
| $\Phi_{31} : (h_{3,1}, h_{3,1})$ | $\to (2,2)$ | $\Phi_{15} : (h_{1,5}, h_{1,5})$ | $\to (2,2)$ |

(2.1)

Here and in what follows, we use the conventions

$$c = 1 - 6\left(\beta - \frac{1}{\beta}\right)^2, \;\; \beta \le 1, \tag{2.2}$$

and

$$h_{rs} = \left(\frac{s\beta}{2} - \frac{r}{2\beta}\right)^2 - \left(\frac{\beta}{2} - \frac{1}{2\beta}\right)^2. \tag{2.3}$$

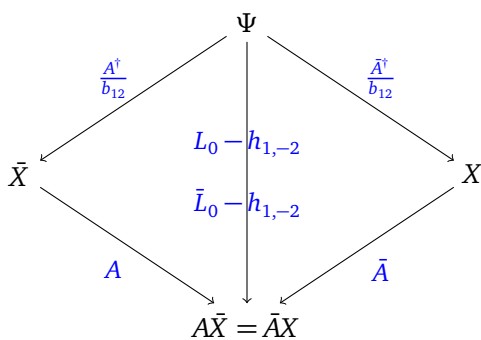

Figure 1: A diamond structure under $\mathcal{V}$ at generic $c$

The value $c = 0$ corresponds to $\beta = \sqrt{\frac{2}{3}}$. All the derivative indicated by prime $'$ are derivatives with respect to $c$.

At generic $c$, the fields $X, \bar{X}$ in (2.1) correspond to the spin-2 4-leg ("hulls") operator in Potts model and the spin-2 2-leg operator in $O(n)$ loop model, respectively. It has been shown recently in [1,3,4] that these fields are involved in a diamond structure under the action of $\mathcal{V}$ at generic $c$. See fig. 1. [1]

Here $\Psi$ is a field of dimensions $(h_{1,-2}, h_{1,-2})$ with $A^\dagger \Psi = L_2 \Psi$. Note that the level-2 null descendants of the primary fields $X, \bar{X}$ coincide: $A\bar{X} = \bar{A}X$, where

$$A = L_{-2} - \frac{3}{2 + 4h_{1,2}(c)} L_{-1}^2, \tag{2.4}$$

and $(\Psi, A\bar{X})$ form a rank-2 Jordan block for $L_0, \bar{L}_0$.

The two-point functions of the logarithmic pair $(\Psi, A\bar{X})$ at generic $c$ are given by [2]

$$\langle \Psi(z,\bar{z}) \Psi(0,\bar{0}) \rangle = \frac{-2b_{12}(c)\ln(z\bar{z}) + \lambda(c)}{(z\bar{z})^{2h_{-1,2}(c)}}, \tag{2.5a}$$

$$\langle \Psi(z,\bar{z}) A\bar{X}(0,\bar{0}) \rangle = \frac{b_{12}(c)}{(z\bar{z})^{2h_{-1,2}(c)}}, \tag{2.5b}$$

$$\langle A\bar{X}(z,\bar{z}) A\bar{X}(0,\bar{0}) \rangle = 0, \tag{2.5c}$$

where $\lambda(c)$ is a unimportant constant (which can be modified, as usual with logarithmic pairs, by a redefinition $\Psi \mapsto \Psi + \text{const}.\bar{A}X$). Note that the above structure appear in both Potts and $O(n)$ models and the corresponding logarithmic couplings are given by[3]:

$$b_{12}^{\text{Potts}}(c) = 2 - \frac{2}{\beta^4} + \frac{4}{\beta^2} - 4\beta^2, \tag{2.6a}$$

$$b_{12}^{O(n)}(c) = 2 + \frac{1}{\beta^6} - \frac{2}{\beta^4} - \frac{1}{\beta^2}, \tag{2.6b}$$

where $\beta$ is given in (2.2), and the top field $\Psi$ in fig. 1 should be labeled $\Psi^{\text{Potts}}$ and $\Psi^{O(n)}$ respectively in the two cases. Below we will use $b_{12}$ and $\Psi$ as common notation for both cases and make the distinction when necessary.

---

[1] we follow the notation in [4]

[2] The normalization of the $\Psi$ is fixed by the relation $A^\dagger \Psi = b_{12}\bar{X}$ and that the constant in the two-point function of $\bar{X}$ is normalized to 1.

[3] See section 6 of [4] for the derivation of the logarithmic coupling for Potts ($b_{12}^{\text{Potts}}$ is given in eq. (94) in that reference). See also eqs. (2.34) and (2.36) of [1] for the logarithmic couplings for Potts and $O(n)$. The definitions of logarithmic couplings in [4] and [1] are different by an overall function of $\beta$. Here we follow the definition of [4].

We now consider the OPE at generic $c$ of two primary fields which for simplicity are taken to be diagonal with dimensions $\Delta = \bar{\Delta}$. At generic $c$, we assume this OPE is given by the following generic form (factoring out $(z\bar{z})^{-2\Delta}$)[4]:

$$
\begin{aligned}
\Phi_\Delta(z,\bar{z}) \times \Phi_\Delta(0,0) \sim{}& 1 + \frac{2\Delta(c)}{c}\big(z^2 T + \bar{z}^2 \bar{T}\big) + \frac{4\Delta(c)^2}{c^2}(z\bar{z})^2 T\bar{T} + \dots \\
&+ \sqrt{\mathcal{A}(c)}\bigg\{(z\bar{z})^{h_{1,2}(c)}\Big(\bar{z}^2 \bar{X} + z^2 X + \frac{z\bar{z}^2}{2}\partial\bar{X} + \frac{z^2\bar{z}}{2}\bar{\partial}X + \alpha(c)(z\bar{z})^2(\partial^2\bar{X} + \bar{\partial}^2 X)\Big) \\
&+ g(c)(z\bar{z})^{h_{-1,2}(c)}\big(\Psi + \ln(z\bar{z})A\bar{X}\big) + \dots\bigg\} + \sqrt{\mathcal{R}(c)}(z\bar{z})^{h_\Phi(c)}\Phi + \dots.
\end{aligned}
$$
(2.7)

Here we use $\Phi$ as a generic notation for $\Phi_{31}$ in Potts and $\Phi_{15}$ in $O(n)$, and similarly with $\Psi$ for $\Psi^{\text{Potts}}$ and $\Psi^{O(n)}$ as mentioned above (their dependency upon $c$ is thus not mentioned explicitly). The conformal weight $\Delta(c)$ is a priori generic, although we will in practice restrict to the spin (order operator) field $\big(h_{\frac{1}{2},0}, h_{\frac{1}{2},0}\big)$ in the Potts model (see appendix B) [5], or the spin field (one-leg operator) $\big(h_{0,\frac{1}{2}}, h_{0,\frac{1}{2}}\big)$ in the $O(n)$ model, where results about four-point functions obtained using the bootstrap method confirm the validity of this expansion. The coefficients $\mathcal{A}$ and $\mathcal{R}$ and $g$ are in principle determined by solving the bootstrap. In particular, $\mathcal{A}$ represents the amplitude of $X, \bar{X}$ appearing in the four-point function $\langle \Phi_\Delta \Phi_\Delta \Phi_\Delta \Phi_\Delta \rangle$ which can be determined from numerical bootstrap, and $\mathcal{R}$ indicate a recursion resulting from degeneracy in the model. Finally, the value $\alpha(c) = \frac{1+h_{1,2}}{4(1+2h_{1,2})}$ follows from conformal invariance [4].

Our strategy will be in particular to demand that the OPE (2.7) has a finite limit as $c \to 0$. Although an old idea, this strategy has so far only been applied to the terms in $z^2, \bar{z}^2$, leading to the construction of $t$ – the logarithmic partner of $T$. Our analysis will go much further, thanks to the recently discovered existence of a rank-two Jordan block at generic $c$ [1, 3, 4] (leading to the last line in (2.7)), combined with the recently obtained results of the couplings $\mathcal{A}, \mathcal{R}$ for at least some fields $\Phi_\Delta$ using the bootstrap [5].

## 2.2 Rank-2 Jordan block of $(t, T)$

We start by reviewing the construction of the logarithmic partner $t$ of the stress-energy tensor $T$ at $c = 0$. This field appears as a combination of the field $X$ and $T$ as follows.

Consider in (2.7) the terms:

$$
\frac{2\Delta(c)}{c}z^2 T + \sqrt{\mathcal{A}(c)}(z\bar{z})^{h_{1,2}(c)}z^2 X,
$$
(2.8)

and similarly for $\bar{z}$. Clearly the coefficient $\frac{1}{c}$ for $T$ is divergent as $c \to 0$ which is usually referred to as the "$c \to 0$ catastrophe" [9]. The question is, how can one - formally at least - make the OPE finite. The solution is to introduce the combination:[5]

$$
t = \frac{b}{\Delta(c)}\Big(\sqrt{\mathcal{A}(c)}X + \frac{2\Delta(c)}{c}T\Big),
$$
(2.9)

(here, $t, X, T$ depend on $c$ although we do not mention it explicitly) where $b$ is given by

$$
b = -\frac{1}{2h'_{1,2}} = -5,
$$
(2.10)

---

[4]Note that the dependence of $\Delta = \Delta(c)$ in this factor should also be taken into account when analyzing the $c \to 0$ limit. However this only gives extra terms which vanish due to the singularity cancellation conditions below. Therefore, for simplicity, we will ignore this dependence below.

[5]Note that this definition appears to be dimensionally problematic for generic $c$ since $X$ and $T$ do not have the same dimension at $c \neq 0$. Instead one should write $t = \frac{b}{\Delta(c)}\big(\sqrt{\mathcal{A}(c)}X\mu^{-2h_{1,2}(c)} + \frac{2\Delta(c)}{c}T\big)$ with some scale $\mu$ which we have suppressed throughout the paper. See [21] for more details. Similar remarks apply to the definition of $\Phi_2$ in (2.26) below.

with $h'_{1,2} = \frac{dh_{1,2}}{dc}|_{c=0}$, and to postulate that, as $c \to 0$, the combination (2.9) becomes a genuine field in the $c = 0$ theory. If this is the case, we see indeed that, as $c \to 0$, (2.8) is now finite and reads:

$$\frac{\Delta}{b} z^2 \big( t + T \ln(z\bar{z}) \big), \tag{2.11}$$

where $\Delta = \Delta(c = 0)$. With the definition (2.9), we can calculate the two-point function of $t$:

$$\langle t(z,\bar{z}) t(0,\bar{0}) \rangle = \frac{b^2}{\Delta^2} \frac{\mathcal{A}(c)\big( 1 - 2h'_{1,2} c \ln(z\bar{z}) \big) + \frac{2\Delta^2}{c} + \dots}{z^4}. \tag{2.12}$$

In order for the two-point function (2.12) to be finite at $c = 0$, the amplitude $\mathcal{A}$ needs to have the following behavior as $c \to 0$:

$$\mathcal{A}(c) = -\frac{2\Delta^2}{c} + \kappa + \mathcal{O}(c). \tag{2.13}$$

Were this not to hold, the construction of $t$ - and in fact, the whole argument trying to salvage finite OPEs at $c = 0$ - would fail. Luckily, this property is known to hold, at least in the case of the Potts model - see appendix B for further detail. We then have the following two-point functions at $c = 0$:

$$\langle t(z,\bar{z}) t(0,\bar{0}) \rangle = \frac{-2b \ln(z\bar{z}) + \theta}{z^4}, \tag{2.14a}$$

$$\langle t(z,\bar{z}) T(0) \rangle = \frac{b}{z^4}, \tag{2.14b}$$

$$\langle T(z) T(0) \rangle = 0, \tag{2.14c}$$

where

$$\theta = \frac{b^2}{\Delta^2} (\kappa + 4\Delta\Delta'). \tag{2.15}$$

From (2.14) we see that the fields $(t, T)$ form a rank-2 Jordan block. Note that $\theta$ is not uniquely defined, as the algebraic definition of $t$ "on top of the Jordan block" allows a redefinition $t \to t + \text{const}.T$.

The remaining terms in the second line of the OPE (2.7) now become:

$$z\bar{z}^2 \frac{\Delta}{2b} \partial \bar{t} + z^2 \bar{z} \frac{\Delta}{2b} \bar{\partial} t + (z\bar{z})^2 \frac{\Delta}{4b} \big( \partial^2 \bar{t} + \bar{\partial}^2 t \big), \tag{2.16}$$

where we have used that $\partial \bar{T} = \bar{\partial} T = 0$ and $\alpha(c = 0) = \frac{1}{4}$.

Using (2.9) and (2.13), we can write, to leading order as $c \to 0$

$$T \overset{c \to 0}{=} -\sqrt{-\frac{c}{2}} X + \frac{c}{2b} t. \tag{2.17}$$

Now, for $t$ to be well-defined in the $c = 0$ CFT, it must have finite correlation functions, just like we saw above in the case of the two-point functions. [6] Assuming this is true indeed, then an insertion of $T(z)$ in correlation functions at $c = 0$ can be replaced by (the limit of) $-\sqrt{-\frac{c}{2}} X$ as long as it comes with a $\mathcal{O}(1)$ coefficient since [7]

$$\langle T(z) \dots \rangle \overset{c \to 0}{=} \frac{c}{2b} \langle t(z,\bar{z}) \dots \rangle - \langle \sqrt{-\frac{c}{2}} X(z,\bar{z}) \dots \rangle \overset{c \to 0}{=} -\langle \sqrt{-\frac{c}{2}} X(z,\bar{z}) \dots \rangle. \tag{2.18}$$

---

[6]The finiteness of three-point functions involving $t$ was considered in [18].

[7]Here we have assumed that ... involves only operators that have a well-defined $c \to 0$ limit, which means that their correlation functions are finite at $c = 0$ and therefore $\langle t(z,\bar{z}) \dots \rangle$ in eq. (2.18) is a finite quantity at $c = 0$.

This allows us to make the following operator identification at $c = 0$:

$$T \overset{c \to 0}{=} -\sqrt{-\frac{c}{2}} X \,, \tag{2.19}$$

where by the right-hand side we of course mean the limit of this expression as $c \to 0$. Note that although $X(z, \bar{z})$ is non-chiral at generic $c$, one has:

$$\langle \bar{\partial} X(z, \bar{z}) \bar{\partial} X(0, \bar{0}) \rangle = \frac{-2h_{12}(c)(2h_{12}(c) + 1)}{z^{2h_{1,-2}(c)} \bar{z}^{2h_{1,2}(c) + 2}} = \mathcal{O}(c) \,, \tag{2.20}$$

and therefore

$$-\sqrt{-\frac{c}{2}} \bar{\partial} X \overset{c \to 0}{=} 0 \,, \tag{2.21}$$

so the identification (2.19) is consistent with that $T$ is chiral at $c = 0$. However $t$ is non-chiral, namely $\bar{\partial} t \neq 0$ due to the $1/\sqrt{c}$ factor in front of $X$ in in (2.9). For example, we have

$$\langle \bar{\partial} t(z, \bar{z}) \bar{\partial} t(0, 0) \rangle = \frac{-2b}{z^4 \bar{z}^2} \,. \tag{2.22}$$

The physics of the identification in (2.19) would certainly deserve more discussion than we can provide here. We note that it is quite natural from the lattice point of view. Indeed, it is known that the states in the transfer matrix for the dense loop (Potts) model corresponding to the left and right hand sides of (2.19) become identical [10, 11]. This is because the rank-two Jordan block appears as $T$ and $X$ (with weights $(h_{1,-2}, h_{1,2})$) mix, and it is a standard property of Jordan blocks that when the degeneracy point is approached by taking the limit of a two by two matrix with slightly different eigenvalues, the two eigendirections coincide. In turn, by state-operator correspondence, the two operators should coincide - note that the numerical factors $\sqrt{-\frac{c}{2}}$ in (2.19) are there to ensure coincidence of normalizations, $X$ being by convention normalized as an ordinary CFT field.

The above computation is identical for Potts and $O(n)$ at $c = 0$ since the field $X$ appear in both models and the leading behavior of the amplitude $\mathcal{A}$ at $c = 0$ is completely determined by the condition (2.13) which leads to the common expression of the field $t$ at $c = 0$:

$$t = b\sqrt{-\frac{2}{c}} X + \frac{2b}{c} T \,. \tag{2.23}$$

A difference between the two models may arise in the $\mathcal{O}(1)$ term of the expansion (2.13) which gives

$$\begin{aligned} t &= b\sqrt{-\frac{2}{c}} X + \frac{2b}{c} T - \text{const.} \sqrt{-\frac{c}{2}} X \\ &= t + \text{const.} T \,, \end{aligned} \tag{2.24}$$

where we have used the identification (2.19) for the second line. As already discussed, this does not modify the rank-2 Jordan block structure.

## 2.3 Rank-3 Jordan block involving $T\bar{T}$

Let us now proceed with trying to render finite the remaining terms in the OPE (2.7) as $c \to 0$. We thus consider the combination (where again the fields also depend on $c$ but we do not indicate this explicitly)

$$(z\bar{z})^2 \frac{4\Delta(c)^2}{c^2} T\bar{T} + (z\bar{z})^{h_\Phi(c)} \sqrt{\mathcal{R}(c)} \Phi + (z\bar{z})^{h_{-1,2}(c)} g(c) \sqrt{\mathcal{A}(c)} \big( \Psi + \ln(z\bar{z}) A\bar{X} \big) \,. \tag{2.25}$$

The field $T\bar{T}$ comes with a problematic divergent coefficient $\sim c^{-2}$. The resolution here is similar to the previous subsection. We define a field

$$\Phi_2 = -\frac{1}{4h_\Phi' \Delta(c)^2}\Big(\frac{4\Delta(c)^2}{c^2}T\bar{T} + \sqrt{\mathcal{R}(c)}\Phi + g(c)\sqrt{\mathcal{A}(c)}\Psi\Big), \tag{2.26}$$

with $h_\Phi' = \frac{dh_\Phi}{dc}|_{c=0}$. Computing the two-point function of $\Phi_2$ we first find

$$\langle \Phi_2(z,\bar{z})\Phi_2(0,\bar{0})\rangle = \frac{1}{16h_\Phi'^2\Delta^4}\frac{\frac{4\Delta^4}{c^2}+\mathcal{R}(c)+\dots}{(z\bar{z})^4}, \tag{2.27}$$

where $\dots$ is subleading. This forces $\mathcal{R}(c)$ to have the behavior

$$\mathcal{R}(c) = -\frac{4\Delta^4}{c^2}+\frac{r_1}{c}+r_0+\mathcal{O}(c), \tag{2.28}$$

in order to cancel the divergence in the numerator at $c=0$. Using (2.28), the two-point function (2.27) further becomes

$$\langle \Phi_2(z,\bar{z})\Phi_2(0,\bar{0})\rangle = \frac{1}{4h_\Phi'^2\Delta^2}\frac{\frac{b_{12}g^2+2h_\Phi'\Delta^2}{c}\ln(z\bar{z})+\frac{r_1-2\lambda g^2\Delta^2+16\Delta^3\Delta'}{4\Delta^2 c}+\dots}{(z\bar{z})^4}, \tag{2.29}$$

where $g, b_{12}, \lambda$ denote the values of $g(c), b_{12}(c), \lambda(c)$ at $c=0$ (recall their definitions at generic $c$ in (2.7),(2.5a)), and $\dots$ is of $\mathcal{O}(1)$. So one needs to further require

$$g^2 = -\frac{2h_\Phi'\Delta^2}{b_{12}}, \quad r_1-2\lambda g^2\Delta^2+16\Delta^3\Delta' = 0. \tag{2.30}$$

The conditions (2.28) and (2.30) can in fact be checked to satisfy in special cases as we will see in appendix B.

We now further define the fields:

$$\Phi_1 = \frac{T\bar{T}}{c}+\frac{1}{2\sqrt{-2b_{12}h_\Phi'}}\sqrt{-\frac{2}{c}}A\bar{X}+\frac{h_{-1,2}'-h_\Phi'}{\sqrt{-2b_{12}h_\Phi'}}\sqrt{-\frac{c}{2}}\Psi+f_1 T\bar{T}-f_2\sqrt{-\frac{c}{2}}A\bar{X}, \tag{2.31a}$$

$$\Phi_0 = h_\Phi' T\bar{T}+\frac{2h_{-1,2}'}{\sqrt{-2b_{12}h_\Phi'}}\sqrt{-\frac{c}{2}}A\bar{X}, \tag{2.31b}$$

where the coefficients $f_1, f_2$ depend on the dimension $\Delta$ and are not characteristic of the logarithmic structure we are investigating, as will become more clear below. The specific expressions will be given in appendix B. Using the definitions of $\Phi_{2,1,0}$ in (2.26), (2.31) and the conditions (2.28) and (2.30), the OPE (2.25) then takes the following form in the limit $c \to 0$:

$$-(z\bar{z})^2 4h_\Phi'\Delta^2\Big(\Phi_2+\ln(z\bar{z})\Phi_1+\frac{1}{2}\ln^2(z\bar{z})\Phi_0\Big), \tag{2.32}$$

where $\Delta = \Delta(c=0)$. From the form of (2.32), we can already expect the fields $(\Phi_2, \Phi_1, \Phi_0)$ to form a rank-3 Jordan block, although we will check this in more detail later.

Notice that the OPE (2.25) describe both Potts and $O(n)$ at generic $c$. As we have mentioned above, in each case, the field $\Phi$ (and its dimension $h_\Phi$) should be taken as

$$\Phi = \begin{cases} \Phi_{31}, & \text{Potts}, \\ \Phi_{15}, & O(n), \end{cases} \tag{2.33}$$

and the field $\Psi$ in the generic $c$ rank-2 Jordan block is $\Psi^{\text{Potts}}$ and $\Psi^{O(n)}$ respectively. In the two cases, we observe that the generic $c$ "logarithmic couplings" (2.6) take the following values at $c = 0$:

$$b_{12}^{\text{Potts}} = -\frac{1}{2h'_{3,1}} = \frac{5}{6}, \tag{2.34a}$$

$$b_{12}^{O(n)} = -\frac{1}{2h'_{1,5}} = -\frac{5}{8}. \tag{2.34b}$$

Therefore in (2.31) we have $\sqrt{-2b_{12}h'_\Phi} = 1$ for both Potts and $O(n)$. Note also

$$h'_{-1,2} = h'_{1,2} = \frac{1}{2}(h'_{3,1} + h'_{1,5}), \tag{2.35}$$

which leads to:[8]

$$\frac{2}{b} = \frac{1}{b_{12}^{\text{Potts}}} + \frac{1}{b_{12}^{O(n)}}. \tag{2.36}$$

Recall now the identification (2.19). Since this is an operator equality, it means that we can also identify their descendants. We therefore claim that , at $c = 0$ (as usual, we must think of the right hand side in the limit $c \to 0$)

$$T\bar{T} \stackrel{c \to 0}{=} -\sqrt{-\frac{c}{2}}\bar{A}X, \tag{2.37}$$

and write (2.31) as

$$\Phi_1 \stackrel{c \to 0}{=} \frac{T\bar{T}}{c} + \frac{1}{\sqrt{-2c}}A\bar{X} + \left(\frac{1}{2b_{12}} - \frac{1}{2b}\right)\sqrt{-\frac{c}{2}}\Psi + (f_1 + f_2)T\bar{T}, \tag{2.38a}$$

$$\Phi_0 \stackrel{c \to 0}{=} \left(\frac{1}{b} - \frac{1}{2b_{12}}\right)T\bar{T}. \tag{2.38b}$$

It is more natural to choose the normalization through a rescaling:

$$\Phi_i \to \Psi_i = \frac{2bb_{12}}{2b_{12} - b}\Phi_i, \quad i = 0, 1, 2, \tag{2.39}$$

such that the bottom field is

$$\Psi_0 = T\bar{T}. \tag{2.40}$$

It is then straightforward to compute the two-point functions of the fields $(\Psi_2, \Psi_1, \Psi_0)$ using their definitions (eqs. (2.26), (2.38) and (2.39)), which take the standard form for rank-3 Jordan block:

$$\langle \Psi_2(z,\bar{z})\Psi_2(0,\bar{0})\rangle = \frac{a_2 - 2a_1 \ln(z\bar{z}) + 2a_0 \ln^2(z\bar{z})}{(z\bar{z})^4}, \tag{2.41a}$$

$$\langle \Psi_2(z,\bar{z})\Psi_1(0,\bar{0})\rangle = \frac{a_1 - 2a_0 \ln(z\bar{z})}{(z\bar{z})^4}, \tag{2.41b}$$

$$\langle \Psi_1(z,\bar{z})\Psi_1(0,\bar{0})\rangle = \frac{a_0}{(z\bar{z})^4}, \tag{2.41c}$$

$$\langle \Psi_2(z,\bar{z})\Psi_0(0,\bar{0})\rangle = \frac{a_0}{(z\bar{z})^4}, \tag{2.41d}$$

$$\langle \Psi_1(z,\bar{z})\Psi_0(0,\bar{0})\rangle = 0, \tag{2.41e}$$

$$\langle \Psi_0(z,\bar{z})\Psi_0(0,\bar{0})\rangle = 0. \tag{2.41f}$$

---

[8]At $c = 0$ notice that $b_{12}^{\text{Potts}} = b_{poly}$ and $b_{12}^{O(n)} = b_{perco}$ as evident from (2.34). The $b_{poly}$ and $b_{perco}$ are the "$b$-numbers" in the boundary (chiral) cases [9]. The relation $\frac{2}{b} = \frac{1}{b_{poly}} + \frac{1}{perco}$ was first observed in [12].

The parameters $a_1, a_2$ are not intrinsic and depends on the external field - that is, the field $\Phi_\Delta$ in the OPE (2.7) . We discuss them further in appendix B. The parameter $a_0$ on the other hand fully characterizes the logarithmic structure (and is independent of the possible field redefinitions within the block). It is given by:

$$a_0 = \frac{b_{12}^2 b}{2b_{12} - b} = -\frac{25}{48}. \tag{2.42}$$

It is interesting to notice the parameter $a_0$ is the same for Potts and $O(n)$ at $c = 0$, i.e., percolation and polymers: Since the $b_{12}$ in these two cases are related through eq. (2.36), it is easy to verify that the expression (2.42) is invariant under

$$b_{12} \leftrightarrow \left(\frac{2}{b} - \frac{1}{b_{12}}\right)^{-1}, \tag{2.43}$$

so interestingly we observe

$$a_0 = b_{12}^{\text{Potts}} b_{12}^{O(n)} = -\frac{25}{48}. \tag{2.44}$$

Combining (2.11), (2.16) and (2.32), we have the following logarithmic OPE at $c = 0$:

$$
\begin{aligned}
\Phi_\Delta(z,\bar{z})\Phi_\Delta(0,\bar{0}) =& (z\bar{z})^{-2\Delta}\Bigg\{1 + z^2\frac{\Delta}{b}\Big(t(0,\bar{0}) + T(0)\ln(z\bar{z})\Big) + \bar{z}^2\frac{\Delta}{b}\Big(\bar{t}(0,\bar{0}) + \bar{T}(\bar{0})\ln(z\bar{z})\Big) \\
&+ z\bar{z}^2\frac{\Delta}{2b}\partial\bar{t}(0,\bar{0}) + z^2\bar{z}\frac{\Delta}{2b}\bar{\partial}t(0,\bar{0}) + (z\bar{z})^2\frac{\Delta}{4b}\Big(\partial^2\bar{t}(0,\bar{0}) + \bar{\partial}^2 t(0,\bar{0})\Big) \\
&+ (z\bar{z})^2\frac{\Delta^2}{a_0}\Big(\Psi_2(0,\bar{0}) + \ln(z\bar{z})\Psi_1(0,\bar{0}) + \frac{1}{2}\ln^2(z\bar{z})\Psi_0(0,\bar{0})\Big) + \dots\Bigg\}.
\end{aligned}
\tag{2.45}
$$

The logarithmic fields $t, \Psi_2, \Psi_1$ arise from the mixing of the generic $c$ fields and we have seen above from their two-point functions that we have a rank-2 Jordan block at dimension $(0,2)$ or $(2,0)$ as well as a rank-3 Jordan block at dimension $(2,2)$. In the next section, we will analyze the Virasoro structure of these fields in more detail based on conformal invariance and the self-duality of the CFT state space.

# 3 Virasoro structure

While the general form of the OPE together the numerical value of the logarithmic couplings at $c = 0$ are the most natural properties to consider, it is also interesting to study the structure of the corresponding $\mathcal{V}$ modules. This, by itself, is a difficult and intricate question, and we will restrict here to considering the "identity" module (i.e., the fields related with the identity field via the action of $\mathcal{V}$) and to conformal weights $h, \bar{h} \le 2$. We note that an attempt to determine the structure of this module has already appeared in [15]: this is discussed briefly in the conclusion.

## 3.1 Consequences of conformal invariance

Let us now take the OPE (2.45) and focus on the requirement of conformal invariance. To do this, we act $L_{n \ge 0}$ on both side of the OPE. On the left hand side, using the conformal Ward identities, we can obtain the differential operator acting on the OPE. We then apply it on the right hand side and compare the expansion in $z, \bar{z}$ order by order to derive the actions of $L_{n \ge 0}$ on the fields in the identity module.

Let us now state the procedure in more detail. On the left hand side of the OPE (2.45), for the primary field $\Phi_\Delta$, we have

$$[L_n, \Phi_\Delta(z,\bar{z})] = \left(z^{n+1}\frac{\partial}{\partial z} + \Delta(n+1)z^n\right)\Phi_\Delta(z,\bar{z}), \tag{3.1}$$

and

$$[L_n, \Phi_\Delta(z,\bar{z})\Phi_\Delta(0,0)] = [L_n, \Phi_\Delta(z,\bar{z})]\Phi_\Delta(0,0) + \Phi_\Delta(z,\bar{z})[L_n, \Phi_\Delta(0,0)]. \tag{3.2}$$

We then find that the action of $L_n$ are given by

$$\begin{cases} z^{n+1}\frac{\partial}{\partial z} + \Delta(n+1)z^n, & n > 0, \\ z\frac{\partial}{\partial z} + 2\Delta, & n = 0, \end{cases} \tag{3.3}$$

where we have used that $L_{n>0}\Phi(0,0) = 0$ and $L_0\Phi_\Delta(0,0) = \Delta\Phi_\Delta(0,0)$. Now we can apply these differential operators on the right-hand side of the OPE (2.45) and compare the coefficients of $z,\bar{z}$ order by order to obtain the actions of $L_n$ on these fields.

First look at $L_0$ for each order. At $z^2$ and $\bar{z}^2$ we find

$$\begin{aligned} L_0 t = 2t + T, \ & L_0 T = 2T, \\ L_0 \bar{t} = \bar{T}, \ & L_0 \bar{T} = 0, \end{aligned} \tag{3.4}$$

so $(t, T)$ indeed form a rank-2 Jordan block.[9] One can also check the terms of $\bar{\partial}t, \partial\bar{t}, \partial^2\bar{t}, \bar{\partial}^2 t$ and their coefficients are simple consequences of the Virasoro algebra. At $(z\bar{z})^2$ we find

$$\begin{aligned} L_0\Psi_2 &= 2\Psi_2 + \Psi_1, \\ L_0\Psi_1 &= 2\Psi_1 + \Psi_0, \\ L_0\Psi_0 &= 2\Psi_0, \end{aligned} \tag{3.5}$$

i.e., a rank-3 Jordan block of $(\Psi_2, \Psi_1, \Psi_0)$ as expected.

Examining $L_1$, we get

$$\begin{aligned} L_1 t = 0, \ & L_1 T = 0, \ L_1 \bar{t} = 0, \ L_1 \bar{T} = 0, \\ L_1\Psi_2 = 0, \ & L_1\Psi_1 = 0, \ L_1\Psi_0 = 0, \end{aligned} \tag{3.6}$$

and

$$\begin{aligned} L_1\partial\bar{t} = 2\bar{T}, \ & L_1\bar{\partial}t = 0, \\ L_1\partial^2\bar{t} = 2\partial\bar{t}, \ & L_1\bar{\partial}^2 t = 0, \end{aligned} \tag{3.7}$$

where (3.7) can be directly calculated using Virasoro algebra.

Let us finally consider $L_2$. First at $z^2$ and $\bar{z}^2$, we get

$$L_2 t = b\mathbb{I}, \ L_2 T = 0, \ L_2 \bar{t} = 0, \ L_2 \bar{T} = 0. \tag{3.8}$$

Then at $(z\bar{z})^2$. We find

$$L_2\Psi_2 = \frac{a_0}{b}\bar{t} - \frac{a_0}{2b\Delta}\bar{T} \to \frac{a_0}{b}\bar{t}, \ L_2\Psi_1 = \frac{a_0}{b}\bar{T}, \ L_2\Psi_0 = 0, \tag{3.9}$$

where in the first equation we have performed a change of basis for the rank-2 Jordan block of $(t, T)$.[10] Note that the above actions can also be checked explicitly using the expressions (2.26), (2.38) and (2.9) in terms of the fields at generic $c$.

---

[9]The actions on the barred fields $(\bar{t}, \bar{T})$ are similar where $L_0$ is replaced by $\bar{L}_0$.

[10]Note that this change of basis, namely $t \to t + \text{const.}T$ modifies the constant in the two-point function of $t$, but does not modify the Virasoro structure we are investigating.

Using state-operator correspondence, we can now write down Virasoro structure of the CFT state space including the $c = 0$ fields in (2.45). For this purpose, instead of $L_2$, it is better to use the combination of Virasoro generators (2.4) which at $c = 0$ becomes:

$$A = L_{-2} - \frac{3}{2}L_{-1}^2, \quad A^\dagger = L_2 - \frac{3}{2}L_1^2, \tag{3.10}$$

and we have: (we indicate the left and right conformal dimensions of the operator under the corresponding states)

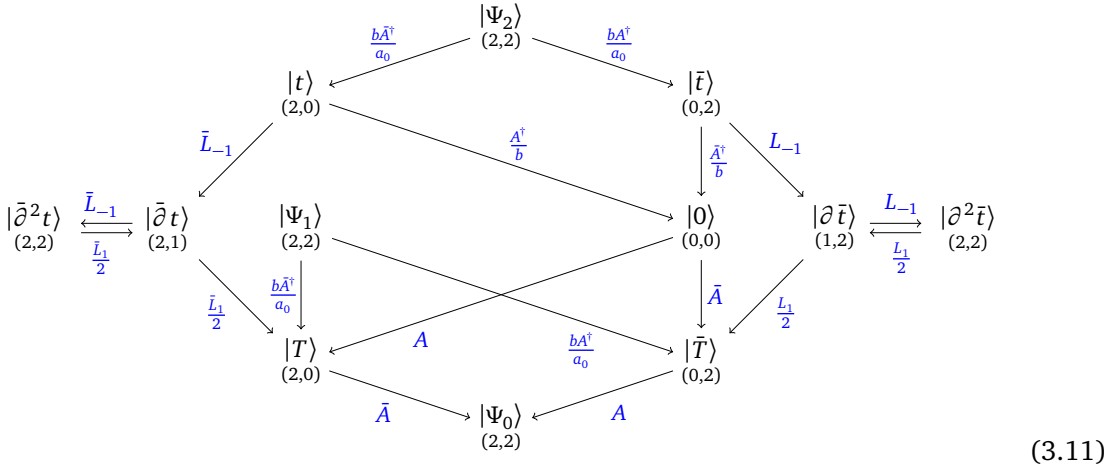

$$\tag{3.11}$$

It is worth stressing that the figure (3.11) results from considering simply the conformal invariance requirement on the logarithmic OPE (2.45) for all the fields up to $h, \bar{h} = 2$. Below in section 3.3, we will add additional arrows to the structure by considering the self-duality of the CFT state space.

Recall that we have

$$A\mathbb{I} = T, \quad \bar{A}\mathbb{I} = \bar{T}, \quad A\bar{T} = \bar{A}T = T\bar{T}, \tag{3.12}$$

where $T\bar{T}$ is our choice of normalization for the bottom field $\Psi_0$. This leads to

$$\bar{A}A\bar{A}^\dagger A^\dagger \Psi_2 = a_0 \Psi_0. \tag{3.13}$$

Here we see that the parameter $a_0$ in (2.42) which characterizes the structure of the logarithmic module is clearly independent of the normalization.[11]

The structure (3.11) cannot be the end of the story because the representing the action of $\mathcal{V}$ is not self-dual, i.e., it is not invariant under reversal of all the arrows. Self-duality of such diagrams is a basic requirement for a physical theory (unitary or not) and guarantees that the Virasoro bilinear form is non-degenerate (see e.g. [13], [15]), or, in physical terms, that there is no field whose two-point function with all other fields in the theory vanishes (of course some of these two-point functions may vanish, for instance $\langle TT \rangle$). See appendix C for an elementary discussion. Hence it is clear that some arrows are missing in (3.11). Meanwhile, a slightly disturbing fact is that the OPE (2.7) for generic $c$ involves six fields

$$\left\{ T\bar{T}, \partial^2 \bar{X}, \bar{\partial}^2 X, \Psi, A\bar{X} = \bar{A}X, \Phi \right\}, \tag{3.14}$$

whose dimensions coincide at $c = 0$ with $(h, \bar{h}) = (2, 2)$. However, in the log OPE (2.45) only five fields with dimension $(2, 2)$ appear after mixing, as can also be seen on the diagram (3.11).

Next, we will study the action of $\mathcal{V}$ further and explore how to make the diagram (3.11) self-dual, and in the meantime introduce a "sixth field" to complete the structure. To start, we calculate the Gram matrix for the states of interest.

---

[11]By normalization, we mean choosing an overall factor $\lambda$ for the fields $\lambda\Phi_i$, $i = 0, 1, 2$. Eq. (3.13) is clearly independent of such a factor.

## 3.2 Gram matrix of $(t, T)$ and $(\Psi_2, \Psi_1, \Psi_0)$

To proceed, we now calculate the Gram matrix involving the logarithmic fields in the rank-2 and rank-3 Jordan blocks of sections 2.2 and 2.3. To do this, one first need to define the bra $\langle \psi |$ for a field $\psi(z, \bar{z})$ :

$$\langle \psi | = \lim_{w, \bar{w} \to 0} \langle 0 | \tilde{\psi}(w, \bar{w}) . \tag{3.15}$$

Here $\tilde{\psi}(w, \bar{w})$ is the transformation of the field $\psi(z, \bar{z})$ under inversion

$$w = \frac{1}{z}, \quad \bar{w} = \frac{1}{\bar{z}} . \tag{3.16}$$

Consider now a generic rank-2 Jordan block $(\psi_1, \psi_0)$ with dimensions $(h, \bar{h})$, under $z \to w(z), \bar{z} \to \bar{w}(\bar{z})$ they transform as

$$
\begin{aligned}
\begin{bmatrix} \tilde{\psi}_1(w, \bar{w}) \\ \tilde{\psi}_2(w, \bar{w}) \end{bmatrix} &= \left(\frac{dz}{dw}\right)^{\begin{bmatrix} h & 1 \\ 0 & h \end{bmatrix}} \left(\frac{d\bar{z}}{d\bar{w}}\right)^{\begin{bmatrix} \bar{h} & 1 \\ 0 & \bar{h} \end{bmatrix}} \begin{bmatrix} \psi_1(z, \bar{z}) \\ \psi_2(z, \bar{z}) \end{bmatrix} \\
&= \left(\frac{dz}{dw}\right)^h \left(\frac{d\bar{z}}{d\bar{w}}\right)^{\bar{h}} \begin{bmatrix} 1 & \ln\left(\frac{dz}{dw} \frac{d\bar{z}}{d\bar{w}}\right) \\ 0 & 1 \end{bmatrix} \begin{bmatrix} \psi_1(z, \bar{z}) \\ \psi_2(z, \bar{z}) \end{bmatrix} .
\end{aligned}
\tag{3.17}
$$

Applying to the $(t, T)$ pair with $(h, \bar{h}) = (2, 0)$ and take into account (3.16), we find

$$\langle t | = \lim_{w, \bar{w} \to 0} \langle 0 | \tilde{t}(w, \bar{w}) = \lim_{z, \bar{z} \to \infty} z^4 \langle 0 | \big( t(z, \bar{z}) + 2\ln(z\bar{z}) T(z) \big) , \tag{3.18a}$$

$$\langle T | = \lim_{w \to 0} \langle 0 | \tilde{T}(w) = \lim_{z \to \infty} z^4 \langle 0 | T(z) . \tag{3.18b}$$

Recall the kets $| \, \rangle$ are defined in the usual way

$$|t\rangle = t(0, 0)|0\rangle , \tag{3.19a}$$

$$|T\rangle = T(0)|0\rangle . \tag{3.19b}$$

It is then straightforward to calculate the gram matrix in the basis $\big(|t\rangle, |T\rangle\big)$ using the two-point functions (2.14) and find

$$\begin{pmatrix} \langle t|t\rangle & \langle t|T\rangle \\ \langle T|t\rangle & \langle T|T\rangle \end{pmatrix} = \begin{pmatrix} \theta & b \\ b & 0 \end{pmatrix} . \tag{3.20}$$

Note that the above is done exactly at $c = 0$ using the two-point functions we obtained in section 2.2. One can also consider the state $|t_c\rangle$ defined at generic $c$ using the definition of operator $t$ in (2.9). Taking the limit $c \to 0$ gives the same result as (3.20). This agrees with our expectation that the CFT state space evolves smoothly as we tune the parameter ($Q$ for Potts and $n$ for $O(n)$) to the $c = 0$ theory.

The case with the rank-3 Jordan block is a straightforward generalization which we do not repeat. Using the two-point functions (2.41), we get the Gram matrix

$$\begin{pmatrix} \langle \Psi_2|\Psi_2\rangle & \langle \Psi_2|\Psi_1\rangle & \langle \Psi_2|\Psi_0\rangle \\ \langle \Psi_1|\Psi_2\rangle & \langle \Psi_1|\Psi_1\rangle & \langle \Psi_1|\Psi_0\rangle \\ \langle \Psi_0|\Psi_2\rangle & \langle \Psi_0|\Psi_1\rangle & \langle \Psi_0|\Psi_0\rangle \end{pmatrix} = \begin{pmatrix} a_2 & a_1 & a_0 \\ a_1 & a_0 & 0 \\ a_0 & 0 & 0 \end{pmatrix} . \tag{3.21}$$

### 3.2.1 Leading terms in the identity conformal block

With the logarithmic OPE in (2.45) and the Gram matrix in the previous section, we can now construct the leading terms of the logarithmic conformal block of the identity module at $c = 0$.

Take the four-point function of identical primary operators $\Phi_\Delta$ placed at $(\infty, 1, z, 0)$:

$$\langle \Phi_\Delta(\infty, \infty)\Phi_\Delta(1,1)\Phi_\Delta(z,\bar{z})\Phi_\Delta(0,0)\rangle = \lim_{z_1,\bar{z}_1 \to \infty}(z_1\bar{z}_1)^{2\Delta}\langle \Phi_\Delta(z_1,\bar{z}_1)\Phi_\Delta(1,1)\Phi_\Delta(z,\bar{z})\Phi_\Delta(0,0)\rangle$$

$$= \sum_{\{\psi\}}\langle \Phi_\Delta|\Phi_\Delta(1,1)|\psi\rangle G^{-1}\langle \psi|\Phi_\Delta(z,\bar{z})|\Phi_\Delta\rangle, \qquad (3.22)$$

where in the last line we have inserted a complete set of CFT states $\{\psi\}$. Take now the subset of states belonging to the logarithmic identity module at $c = 0$. We have the logarithmic Virasoro conformal block of the identity module

$$\mathcal{F}_{\mathbb{I}}^{\log}(z,\bar{z}) \equiv \sum_{\{\psi\}_{\mathbb{I}}}\langle \Phi_\Delta|\Phi_\Delta(1,1)|\psi\rangle G^{-1}\langle \psi|\Phi_\Delta(z,\bar{z})|\Phi_\Delta\rangle, \qquad (3.23)$$

where the sum now goes over states belong to the identity modules appearing in the logarithmic OPE (2.45):

$$\{\psi\}_{\mathbb{I}}: \ \mathbb{I}, \ T, \ \bar{T}, \ t, \ \bar{t}, \ \partial\bar{t}, \ \bar{\partial}t, \ \partial^2\bar{t}, \ \bar{\partial}^2 t, \ \Psi_0, \ \Psi_1, \ \Psi_2, \ \ldots, \qquad (3.24)$$

and $G^{-1}$ is the inverse Gram matrix which can be obtained using (3.20) and (3.21) and the usual Virasoro algebra. The functions $\langle \Phi_\Delta|\Phi_\Delta(1,1)|\psi\rangle$ and $\langle \psi|\Phi_\Delta(z,\bar{z})|\Phi_\Delta\rangle$ are defined by

$$\langle \Phi_\Delta|\Phi_\Delta(1,1)|\psi\rangle \ = \ \lim_{z_1,\bar{z}_1 \to \infty}\langle \Phi_\Delta(z_1,\bar{z}_1)\Phi_\Delta(1,1)\psi(0,0)\rangle, \qquad (3.25)$$

$$\langle \psi|\Phi_\Delta(z,\bar{z})|\Phi_\Delta\rangle \ = \ \lim_{w,\bar{w} \to 0}\langle \tilde{\psi}(w,\bar{w})\Phi_\Delta(z,\bar{z})\Phi_\Delta(0,0)\rangle, \qquad (3.26)$$

where $\tilde{\psi}$ is the field transformed under inversion (3.16). The three-point functions $\langle \Phi_\Delta \Phi_\Delta \psi\rangle$ can be fixed by the Ward identities, the OPE (2.45) and the two-point functions (2.12), (2.41). We give their explicit expressions in appendix A. Plugging the expressions into (3.22), we obtain the leading terms of the logarithmic conformal block of identity module:

$$\mathcal{F}_{\mathbb{I}}^{\log}(z,\bar{z}) = (z\bar{z})^{-2\Delta}\Big\{1 + \frac{\Delta^2}{b^2}(z^2 + \bar{z}^2)\big(\theta + b\ln(z\bar{z})\big) + \frac{\Delta^2}{2b}(z\bar{z}^2 + z^2\bar{z}) + \frac{\Delta^2}{2b}(z\bar{z})^2$$

$$+ \frac{\Delta^4}{a_0^2}(z\bar{z})^2\Big[a_2 + a_1\ln(z\bar{z}) + \frac{a_0}{2}\ln^2(z\bar{z})\Big] + \ldots\Big\}. \qquad (3.27)$$

Note that in the Potts model, the identity modules appears in the geometrical four-point function $P_{aabb}$ in the Fortuin-Kasteleyn cluster formulation, as studied in [5, 16, 17]. In appendix B, we will check the expression (3.27) by taking directly the $c \to 0$ limit of the four-point function $P_{aabb}$.

## 3.3 A self-dual structure

Let us now go back to the structure (3.11) which we have deduced from the requirement of conformal invariance.

It is clear that the structure is incomplete: Since we have

$$\frac{b}{a_0}\bar{A}^\dagger|\Psi_1\rangle = |T\rangle, \qquad (3.28)$$

and $\langle t|T\rangle = b$ from (3.20), this means

$$\langle t|\bar{A}^\dagger|\Psi_1\rangle = \langle \bar{A}t|\Psi_1\rangle = a_0. \qquad (3.29)$$

Therefore, we should also have arrows "going out" from $|t\rangle$ towards states with dimensions $(2, 2)$, which are missing in the diagram (3.11). To clarify these actions, note that in (3.29), there is an ambiguity: Due to the action $L_1 \Psi_1 = 0$, the argument from (3.28) to (3.29) actually works for arbitrary combination of Virasoro operators $\mathcal{P}(\bar{A}, \bar{L}_{-1}^2) = \bar{A} + \alpha \bar{L}_{-1}^2$ so instead of (3.29), we write

$$\langle \mathcal{P}(\bar{A}, \bar{L}_{-1}^2) t | \Psi_1 \rangle = a_0 \,, \tag{3.30}$$

and similarly

$$\langle \mathcal{P}(A, L_{-1}^2) \bar{t} | \Psi_1 \rangle = a_0 \,. \tag{3.31}$$

According to the Gram matrix (3.21), eqs. (3.30), (3.31) seems to suggest that

$$\mathcal{P}(A, L_{-1}^2) | \bar{t} \rangle \,, \; \mathcal{P}(\bar{A}, \bar{L}_{-1}^2) | t \rangle \sim | \Psi_1 \rangle \,, \; | \Psi_2 \rangle \,. \tag{3.32}$$

It is however immediate to see that the second option $|\Psi_2\rangle$ violates self-duality requirement of the structure, since

$$\mathcal{P}(\bar{A}, \bar{L}_{-1}^2) | t \rangle \sim | \Psi_2 \rangle \Longleftrightarrow \langle \Psi_0 | \mathcal{P}(\bar{A}, \bar{L}_{-1}^2) | t \rangle \neq 0 \Longleftrightarrow \langle \mathcal{P}(\bar{A}^\dagger, \bar{L}_1^2) \Psi_0 | t \rangle \neq 0 \,, \tag{3.33}$$

contradicting the actions of $L_2, L_1$ on $\Psi_0$ from (3.6) and (3.9). Therefore it is tempting to claim

$$\mathcal{P}(\bar{A}, \bar{L}_{-1}^2) | t \rangle = \mathcal{P}(A, L_{-1}^2) | \bar{t} \rangle = | \Psi_1 \rangle \,. \tag{3.34}$$

Now let us check if this is the $|\Psi_1\rangle$ we want, using the conformal invariance requirement from section 3.1:

$$L_1 | \Psi_1 \rangle = \bar{L}_1 | \Psi_1 \rangle = 0 \,, \quad L_2 | \Psi_1 \rangle = \frac{a_0}{b} | \bar{T} \rangle \,, \quad \bar{L}_2 | \Psi_1 \rangle = \frac{a_0}{b} | T \rangle \,, \tag{3.35}$$

we first see that by requiring

$$L_1 (A + \alpha L_{-1}^2) | \bar{t} \rangle = \left( [L_1, A] + \alpha \left[ L_1, L_{-1}^2 \right] \right) | \bar{t} \rangle = 2 \alpha L_{-1} | \bar{t} \rangle = 0 \,, \tag{3.36}$$

we fix the coefficient $\alpha = 0$.

$$\mathcal{P}(A, L_{-1}^2) = A = L_{-2} - \frac{3}{2} L_{-1}^2 \,. \tag{3.37}$$

Note it is crucial in the last equality of (3.36) that we are studying a non-chiral case with $L_{-1} | \bar{t} \rangle \neq 0$. The same action on $\bar{A} | t \rangle$ is trivially satisfied since

$$L_1 \bar{A} | t \rangle = \bar{A} L_1 | t \rangle = 0 \,, \tag{3.38}$$

where we have used $L_1 t = 0$ (eq. (3.6)). The first equality in (3.34) now becomes

$$\bar{A} | t \rangle = A | \bar{t} \rangle \,, \tag{3.39}$$

and this condition fixes the value of the parameter $b$ by acting with $L_2$:

$$b | \bar{T} \rangle = \bar{A} L_2 | t \rangle = L_2 \bar{A} | t \rangle = L_2 A | \bar{t} \rangle = [L_2, A] | \bar{t} \rangle = -5 L_0 | \bar{t} \rangle = -5 | \bar{T} \rangle \,, \tag{3.40}$$

so $b = -5$. The fact that the equality (3.39) leads to the value of $b = -5$ was first pointed out in [14] although there, the equality was proposed as an assumption to explain the value of $b$ measured on the lattice by [12]. Here, we see that starting from the generic form of logarithmic OPE (2.45), the equality (3.39) and the value of $b$ is uniquely fixed by requiring conformal invariance and the self-duality of the structure.

Now comparing (3.40) with the action $L_2, \bar{L}_2$ in (3.35), we can define the state

$$|\hat{\Psi}_1\rangle = \frac{a_0}{b^2}|A\bar{t}\rangle = \frac{a_0}{b^2}|\bar{A}t\rangle, \tag{3.41}$$

and

$$|\Psi_1\rangle = |\hat{\Psi}_1\rangle + |\tilde{\Psi}_1\rangle, \tag{3.42}$$

where we have included a term $|\tilde{\Psi}_1\rangle$ since the above identification of $|A\bar{t}\rangle \sim |\Psi_1\rangle$ is checked through the actions of $L_1, L_2$ so they could be different by a state which is annihilated by $L_1, L_2$:

$$L_1|\tilde{\Psi}_1\rangle = L_2|\tilde{\Psi}_1\rangle = 0. \tag{3.43}$$

To finish the structure, let us compute the norm of $|\Psi_1\rangle$ and compare with the Gram matrix (3.21). We have

$$\langle\Psi_1|\Psi_1\rangle = \frac{a_0^2}{b^4}\langle t|\left[\bar{A}^\dagger, \bar{A}\right]|t\rangle + \frac{a_0}{b^2}\left(\langle\bar{A}t|\tilde{\Psi}_1\rangle + \langle\tilde{\Psi}_1|\bar{A}t\rangle\right) + \langle\tilde{\Psi}|\tilde{\Psi}\rangle = \frac{a_0^2}{b^2} + \langle\tilde{\Psi}|\tilde{\Psi}\rangle = a_0, \tag{3.44}$$

which gives

$$\langle\tilde{\Psi}_1|\tilde{\Psi}_1\rangle = a_0\left(1 - \frac{a_0}{b^2}\right). \tag{3.45}$$

In the second to last equality of (3.44), we have used that $\langle\hat{\Psi}_1|\tilde{\Psi}_1\rangle = 0$ which is again justified by self-duality:

$$\langle\hat{\Psi}_1|\tilde{\Psi}_1\rangle \sim \langle A\bar{t}|\tilde{\Psi}_1\rangle = \langle\bar{t}|A^\dagger\tilde{\Psi}_1\rangle = 0, \tag{3.46}$$

since $|\tilde{\Psi}_1\rangle$ is annihilated by $L_1, L_2$.

Let us now draw the final structure:

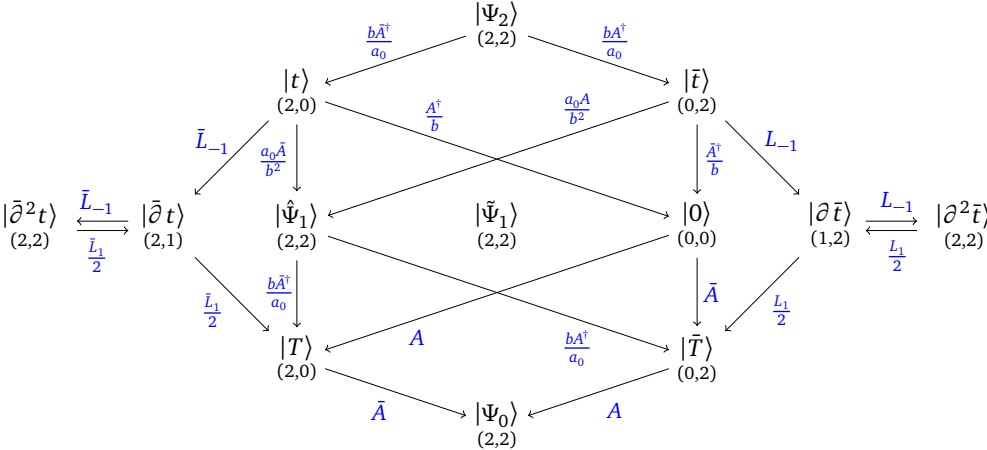

Figure 2: The identity module under $\mathcal{V}$ up to fields with weight $(2, 2)$ at $c = 0$

Note that we have not drawn the action of $L_0, \bar{L}_0$ in figure 2. The state $|\tilde{\Psi}_1\rangle$ is connected to the rest of the depicted states under the action of $L_0, \bar{L}_0$, as will become more clear in the next subsection.

### 3.3.1 The fields $\hat{\Psi}_1$ and $\tilde{\Psi}_1$

We have seen above that to construct a self-dual structure fig. 2, the state $|\Psi_1\rangle$ corresponding to the middle field in the rank-3 Jordan block is necessarily split into two orthogonal parts which we call $|\hat{\Psi}_1\rangle, |\tilde{\Psi}_1\rangle$. In this subsection, we clarify their field content.

First note that according to our definition of $t$ in (2.9), the identity (3.39) is simply

$$A\bar{t} = b\sqrt{-\frac{2}{c}}\bar{A}X + \frac{2b}{c}T\bar{T} = \bar{A}t\,, \tag{3.47}$$

due to $A\bar{X} = \bar{A}X$ from generic $c$ (see fig. 1). Now, taking the definition of the field $\Psi_1$ from eqs. (2.38a) and (2.39), we can write

$$\begin{aligned}
\Psi_1 &\sim \frac{b_{12}}{2b_{12}-b}A\bar{t} + \frac{b-b_{12}}{2b_{12}-b}\sqrt{-\frac{c}{2}}\Psi \\
&= \frac{b_{12}^2}{b(2b_{12}-b)}A\bar{t} + \frac{b-b_{12}}{2b_{12}-b}\Big(\sqrt{-\frac{c}{2}}\Psi + \frac{b_{12}}{b}A\bar{t}\Big)\,,
\end{aligned} \tag{3.48}$$

(where recall $\Psi$ is defined in fig. 1). In the first line of (3.48), we have neglected the terms $\sim T\bar{T}$ which amounts to a change of basis in the rank-3 Jordan block and is unimportant for the logarithmic structure. Under the Virasoro action $L_2$, we find

$$L_2\Psi_1 = \frac{b_{12}^2}{2b_{12}-b}\bar{T} + \frac{b-b_{12}}{2b_{12}-b}b_{12}\Big(\sqrt{-\frac{c}{2}}\bar{X} + \bar{T}\Big) = \frac{a_0}{b}\bar{T}\,, \tag{3.49}$$

where we have used the identification (2.19). Note that due to this identification, the combination in the parenthesis in the second line of (3.48) in fact decouples from the structure under the actions of $L_1, L_2$. Similar computations can be done for $\overline{Vir}$ actions. For this reason, it make sense to define two fields

$$\hat{\Psi}_1 = \frac{b_{12}^2}{b(2b_{12}-b)}A\bar{t} = \frac{a_0}{b^2}A\bar{t} = \frac{a_0}{b^2}\bar{A}t\,, \tag{3.50a}$$

$$\tilde{\Psi}_1 = \frac{b-b_{12}}{2b_{12}-b}\Big(\sqrt{-\frac{c}{2}}\Psi + \frac{b_{12}}{b}A\bar{t}\Big)\,, \tag{3.50b}$$

corresponding to the states $|\hat{\Psi}_1\rangle$ and $|\tilde{\Psi}_1\rangle$ we have seen above, and

$$\Psi_1 = \hat{\Psi}_1 + \tilde{\Psi}_1\,. \tag{3.51}$$

Their two-point functions can be easily extracted from the definition (3.50) and in particular we find

$$\langle\hat{\Psi}_1(z,\bar{z})\tilde{\Psi}_1(0,0)\rangle = 0\,, \tag{3.52}$$

agreeing with the expectation from (3.46).

Note however that the field $\tilde{\Psi}_1$ does not decouple from the rank-3 Jordan block which can be seen for example in the two-point function:

$$\langle\Psi_2(z,\bar{z})\tilde{\Psi}_1(0,\bar{0})\rangle = \frac{2b_{12}^2(b-b_{12})^2}{(2b_{12}-b)^2}\frac{\ln(z\bar{z})}{(z\bar{z})^2}\,. \tag{3.53}$$

Essentially, $\tilde{\Psi}_1$ couples to the structure through the action of $L_0, \bar{L}_0$. It is easy to calculate that:

$$(L_0-2)\hat{\Psi}_1 = \frac{a_0}{b^2}T\bar{T} = -\frac{1}{48}T\bar{T}\,, \tag{3.54a}$$

$$(L_0-2)\tilde{\Psi}_1 = \Big(1-\frac{a_0}{b^2}\Big)T\bar{T} = \frac{49}{48}T\bar{T}\,. \tag{3.54b}$$

It is worth pointing out that the expression (3.53) and the actions (3.54) are also manifestly invariant under (2.43), and this shows the splitting (3.51) is identical for percolation and polymers.

As a final remark, note that definitions of $\hat{\Psi}_1$ and $\tilde{\Psi}_1$ in (3.50) introduce a "sixth field" into the structure fig. 2, thus resolving the "disturbing fact" we have mentioned around eq. (3.14).

### 3.4 Comparison with [1]

Building Jordan blocks by introducing new fields obtained as formal derivatives of primary fields with respect to conformal dimensions is an idea that has been around since the early days of LCFTs. It was explored quite systematically in the recent paper [1], where the authors constructed in particular a rank-3 Jordan block $\left(\widetilde{W}^{\kappa}_{(r,s)}, W^{\kappa}_{(r,s)}, V_{(r,s)}\right)$ for generic values of $c$. While it is believed [3, 4] that there is no rank-3 Jordan block in the Potts or $O(n)$ CFTs at generic $c$, we can nonetheless choose the particular value $c = 0$ in the structure obtained in [1], and compare with fig. 2.

To properly make such comparison, note that the construction in [1] is done by focusing on the null descendant of a primary field at a certain level. In our case, we chose the identity field, which at $c = 0$ obeys:

$$(h_{1,1}, h_{1,1}) = (h_{1,2}, h_{1,2}). \tag{3.55}$$

As a result, its level-2 descendant given by (2.4) – the stress-energy tensor – becomes null. We then use the construction in [1] by taking their $(r, s) = (2, 1)$. [12]

The main characteristic of the structure fig. 2 is the parameter $a_0$ which, as we have seen from (3.13), does not depend on the normalization we choose for the fields $\Psi_i, i = 0, 1, 2$. This is equivalent to the normalization independent structural parameter $\kappa^0_{(2,1)}$ defined in [1] (eq. (2.41) in that reference):

$$\mathcal{L}_{(2,1)}\bar{\mathcal{L}}_{(2,1)}\mathcal{D}\bar{\mathcal{D}}\widetilde{W}^0_{(2,1)} = \kappa^0_{(2,1)}(L_0 - \Delta_{(2,-1)})^2 \widetilde{W}^0_{(2,1)}, \tag{3.56}$$

where $\mathcal{D}$ represents combinations of Virasoro generators $L_{n>0}$ at level 2 and $\mathcal{L}_{(2,1)} = L^2_{-1} - \beta^2 L_{-2}$. At $c = 0$, the parameter $\kappa^0_{(2,1)}$ takes the value

$$\kappa^0_{(2,1)} = -\frac{1}{48}. \tag{3.57}$$

Compare eq. (3.56) with (3.13), and take into consideration of the different normalization of our Virasoro generators from theirs:

$$\mathcal{L}_{(2,1)}\bar{\mathcal{L}}_{(2,1)}\mathcal{D}\bar{\mathcal{D}} = \left(\frac{\beta^4}{4(1-\beta^4)}\right)^2 A\bar{A}A^{\dagger}\bar{A}^{\dagger} = 25 A\bar{A}A^{\dagger}\bar{A}^{\dagger}. \tag{3.58}$$

Eq. (3.57) agrees with our parameter $a_0$ from (2.44).

Recall from section 3.3 that in drawing the self-dual structure fig. 2, we have split the middle field $\Psi_1$ from the rank-3 Jordan block into two parts: $\hat{\Psi}_1$ and $\tilde{\Psi}_1$. The $\tilde{\Psi}_1$ in the middle of the diagram fig. 2 is disconnected from the rest, but is however coupled to the structure through the action of $L_0$. From (3.54) it is clear that the coefficient in the splitting, i.e., $\frac{a_0}{b^2}$ and $1 - \frac{a_0}{b^2}$, are also normalization independent and provide another comparison. The corresponding expressions can be similarly written for the structure of [1]. Their middle field is given by

$$W^0_{(2,1)} = (1 - \kappa^0_{(2,1)})V'_{(2,-1)} + \kappa^0_{(2,1)}\mathcal{L}\bar{\mathcal{L}}V'_{(2,1)}, \tag{3.59}$$

where $V'_{(2,-1)}$ is annihilated by Virasoro generators $L_{n>0}, \bar{L}_{n>0}$ and the construction using the derivative allows one to write:

$$(L_0 - 2)V'_{(2,-1)} = (\bar{L}_0 - 2)V'_{(2,-1)} = V_{(2,-1)}, \tag{3.60a}$$

$$(L_0 - 2)\mathcal{L}\bar{\mathcal{L}}V'_{(2,1)} = \mathcal{L}\bar{\mathcal{L}}(L_0 - 2)V'_{(2,1)} + [L_0, \mathcal{L}]\bar{\mathcal{L}}V'_{(2,1)} = \mathcal{L}\bar{\mathcal{L}}V_{(2,1)} = V_{(2,-1)}. \tag{3.60b}$$

---

[12] Note that our convention for $(r, s)$ is switched from theirs. So our Kac indices $(1, 2)$ as in e.g. (3.55) corresponds to their Kac indices $(2, 1)$. For the rest of this subsection we will follow the convention in [1].

Therefore we see that the field (3.59) indeed is split into two parts with the same coefficients as (3.54) since

$$\kappa^0_{(2,1)} = \frac{a_0}{b^2}.$$ 

(3.61)

The Virasoro structure of [1] can be summarized as following:

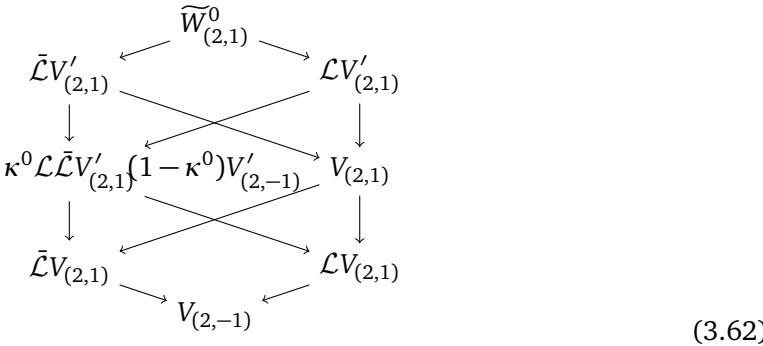

(3.62)

which takes the same form as the "centre" of our structure fig. 2 [13] where the rank-3 Jordan block $(\Psi_2, \Psi_1, \Psi_0)$ is involved.

We emphasize that the construction in [1] holds for the primary field $(h_{12}, h_{12})$ as a starting point, and it is only the coincidence of conformal weights $h_{12} = h_{11} = 0$ at $c = 0$ that makes it relevant to our problem. It is thus not clear to us why the formal structure proposed in [1] should coincide with our (admittedly, obtained much more laboriously) results: the question remains open as to whether this is more than an accident.

## 4   Conclusions

This paper uncovers the non-chiral structure associated with the existence of a logarithmic partner of the stress-energy tensor in percolation and polymers CFTs with $c = 0$. While it is certainly satisfactory to have this finally worked out, it would be very nice to have some independent confirmation based on the analysis of lattice models in the spirit of [11, 15]. We note in this respect that our structure of the identity module (in particular, the existence of a rank-three Jordan block) is compatible with what was proposed in [15]: a more detailed comparison will appear elsewhere.

There are many other possible directions for future work on this difficult problem. One is to compare the structure we have obtained with the replica approach pioneered in [22], [21]. Yet another direction is to revisit the ideas in [9] and try to properly define the action of $t$ modes in the non-chiral case. It would also be interesting to find out how universal the diagram in figure 2 might be, and what the situation is for other $c = 0$ theories. Finally, it is important to emphasize that we have only uncovered the beginning of the identity module structure: what happens for $h, \bar{h} > 2$ remains to be explored. We hope to get back to (some of) these questions in further work.

## Acknowledgements

We thank J.L. Jacobsen, A. Gainutdinov, L. Grans-Samuelsson, L. Liu, R. Nivesvivat, S. Ribault and R. Vasseur for useful discussions, and S. Ribault for comments on the manuscript. We also thank the anonymous referees from SciPost for many comments and suggestions on improving

---

[13]excluding the two sides involving $\partial \bar{t}, \bar{\partial} t, \partial^2 \bar{t}, \bar{\partial}^2 t$

our manuscript. H. S. also thanks A. Gainutdinov and R. Vasseur for an early collaboration on this topic. Our work was supported in part by the advanced ERC grant NuQFT.

# A   Three-point functions of $\langle \Phi_\Delta \Phi_\Delta \psi \rangle$

In section 3.2.1, we have constructed the leading terms in the logarithmic conformal block of the identity module at $c = 0$. To do this, we used the three-point functions of the type $\langle \Phi_\Delta \Phi_\Delta \psi \rangle$ where $\psi$ belongs to a logarithmic multiplet. We now give their explicit expressions.

The position dependence of the three-point functions involving logarithmic fields are fixed by conformal Ward identities. For details see for example [23, 24]. In our case, we focus on $(t, T)$ and $(\Psi_2, \Psi_1, \Psi_0)$. Denoting

$$z_{ij} = z_i - z_j \tag{A.1}$$

the three-point functions are given by

$$\langle \Phi_\Delta(z_1, \bar{z}_1)\Phi_\Delta(z_2, \bar{z}_2)T(z_3)\rangle = \frac{C_{\Phi\Phi T}}{z_{12}^{2\Delta-2}z_{13}^2 z_{23}^2 \bar{z}_{12}^{2\Delta}}, \tag{A.2a}$$

$$\langle \Phi_\Delta(z_1, \bar{z}_1)\Phi_\Delta(z_2, \bar{z}_2)t(z_3, \bar{z}_3)\rangle = \frac{C_{\Phi\Phi t} + C_{\Phi\Phi T}\ln\frac{z_{12}\bar{z}_{12}}{z_{13}\bar{z}_{13}z_{23}\bar{z}_{23}}}{z_{12}^{2\Delta-2}z_{13}^2 z_{23}^2 \bar{z}_{12}^{2\Delta}}, \tag{A.2b}$$

and

$$\langle \Phi_\Delta(z_1, \bar{z}_1)\Phi_\Delta(z_2, \bar{z}_2)\Psi_0(z_3, \bar{z}_3)\rangle = \frac{C_{\Phi\Phi\Psi_0}}{z_{12}^{2\Delta-2}z_{13}^2 z_{23}^2 \bar{z}_{12}^{2\Delta-2}\bar{z}_{13}^2 \bar{z}_{23}^2}, \tag{A.3a}$$

$$\langle \Phi_\Delta(z_1, \bar{z}_1)\Phi_\Delta(z_2, \bar{z}_2)\Psi_1(z_3, \bar{z}_3)\rangle = \frac{C_{\Phi\Phi\Psi_1} + C_{\Phi\Phi\Psi_0}\ln\frac{z_{12}\bar{z}_{12}}{z_{13}\bar{z}_{13}z_{23}\bar{z}_{23}}}{z_{12}^{2\Delta-2}z_{13}^2 z_{23}^2 \bar{z}_{12}^{2\Delta-2}\bar{z}_{13}^2 \bar{z}_{23}^2}, \tag{A.3b}$$

$$\langle \Phi_\Delta(z_1, \bar{z}_1)\Phi_\Delta(z_2, \bar{z}_2)\Psi_2(z_3, \bar{z}_3)\rangle = \frac{C_{\Phi\Phi\Psi_2} + C_{\Phi\Phi\Psi_1}\ln\frac{z_{12}\bar{z}_{12}}{z_{13}\bar{z}_{13}z_{23}\bar{z}_{23}} + \frac{1}{2}C_{\Phi\Phi\Psi_0}\ln^2\frac{z_{12}\bar{z}_{12}}{z_{13}\bar{z}_{13}z_{23}\bar{z}_{23}}}{z_{12}^{2\Delta-2}z_{13}^2 z_{23}^2 \bar{z}_{12}^{2\Delta-2}\bar{z}_{13}^2 \bar{z}_{23}^2}. \tag{A.3c}$$

Inserting the OPE (2.45) and using the two-point functions (2.14) and (2.41), one finds

$$\begin{aligned} C_{\Phi\Phi T} &= \Delta, \quad C_{\Phi\Phi t} = \Delta\frac{\theta}{b}, \\ C_{\Phi\Phi\Psi_0} &= \Delta^2, \quad C_{\Phi\Phi\Psi_1} = \Delta^2\frac{a_1}{a_0}, \quad C_{\Phi\Phi\Psi_2} = \Delta^2\frac{a_2}{a_0}. \end{aligned} \tag{A.4}$$

# B   The $c \to 0$ limit of Potts $P_{aabb}$

As studied in [5, 16, 17], the identity module appears in the Potts geometrical four-point function $P_{aabb}$ where the first two points belong to one Fortuin-Kasteleyn cluster and the last two belong to a different one. At generic $c$, the following combination of conformal blocks enters the four-point function:

$$\langle \Phi_{\frac{1}{2},0}\Phi_{\frac{1}{2},0}\Phi_{\frac{1}{2},0}\Phi_{\frac{1}{2},0}\rangle = \mathcal{F}_{h_{1,1}}(z)\mathcal{F}_{h_{1,1}}(\bar{z}) + \mathcal{R}_{3,1}\mathcal{F}_{h_{3,1}}(z)\mathcal{F}_{h_{3,1}}(\bar{z}) + A_{aabb}(h_{1,2})\mathcal{F}^{\log}_{h_{1,2}}(z, \bar{z}) + \dots, \tag{B.1}$$

where the amplitude for identity $(h_{1,1}, h_{1,1})$ is normalized to 1. The coefficient $\mathcal{R}_{3,1}$ of the field $\Phi_{31}$ appears in the interchiral block [5] of the affine Temperley-Lieb module $\overline{\mathcal{W}}_{0,q^2}$ and is determined from the degeneracy of the field $\Phi_{21}$. (See eq. (4.26) in [5].) $A_{aabb}(h_{1,2})$ represents the amplitude of $X, \bar{X}$ and $\mathcal{F}^{\log}_{h_{1,2}}(z, \bar{z})$ is the logarithmic block obtained in [4].

In the main text, we have studied the logarithmic mixing of the Virasoro modules at $c = 0$ for a generic OPE (2.7) and obtained conditions (2.13), (2.28) and (2.30) which are necessary for canceling the naive divergence at $c = 0$. One can think of this as the condition for the operator $\Phi_\Delta$ in (2.45) to probe the logarithmic structure of the CFT. In the case $\Phi_\Delta = \Phi_{\frac{1}{2},0}$ – the spin operator in Potts model, we can check the logarithmic conformal block (3.27) thus obtained with the direct $c \to 0$ limit of (B.1).

We start with $\Phi_{3,1}$. The recursion $\mathcal{R}_{3,1}$ at generic $c$ for the four-point function (B.1) is given by

$$\mathcal{R}_{3,1}(\beta) = \frac{\Gamma\left(2 - \frac{4}{\beta^2}\right)\Gamma\left(1 - \frac{3}{\beta^2}\right)\Gamma\left(2 - \frac{3}{\beta^2}\right)\Gamma\left(\frac{1}{\beta^2}\right)^2\Gamma\left(\frac{3}{2\beta^2}\right)^4\Gamma\left(\frac{2}{\beta^2}\right)}{\Gamma\left(1 - \frac{2}{\beta^2}\right)\Gamma\left(1 - \frac{3}{2\beta^2}\right)^4\Gamma\left(1 - \frac{1}{\beta^2}\right)^2\Gamma\left(\frac{3}{\beta^2} - 1\right)\Gamma\left(\frac{4}{\beta^2} - 1\right)\Gamma\left(\frac{3}{\beta^2}\right)}, \tag{B.2}$$

and its behavior as $c \to 0$ is

$$\mathcal{R}_{3,1} = \frac{r_2}{c^2} + \frac{r_1}{c} + r_0 + \mathcal{O}(c^1), \tag{B.3}$$

where the coefficients $r_2, r_1, r_0$ can be easily extracted. Expanding the $s$-channel conformal block in $c$ and keep up to $c^2$, we have

$$\mathcal{F}_{h_{3,1}}(z) = z^2\left(1 + h'_{31}c\ln z + \frac{c^2}{2}\left(h''_{31}\ln z + h'^2_{31}\ln^2 z\right) + \dots\right), \tag{B.4}$$

where $h'_{31}, h''_{31}$ denote their values at $c = 0$. The second term in (B.1) is therefore given by

$$\begin{aligned}\mathcal{R}_{3,1}\mathcal{F}_{h_{3,1}}(z)\mathcal{F}_{h_{3,1}}(\bar{z}) = (z\bar{z})^2\bigg(&\frac{r_2}{c^2} + \frac{r_1 + r_2 h'_{3,1}\ln(z\bar{z})}{c} + r_0 \\ &+ h'_{3,1}r_1\ln(z\bar{z}) + \frac{r_2 h''_{3,1}}{2}\ln(z\bar{z}) + \frac{r_2 h'^2_{3,1}}{2}\ln^2(z\bar{z}) + \dots\bigg).\end{aligned} \tag{B.5}$$

Now expanding the $s$-channel conformal block for identity in $c$, we have

$$\mathcal{F}_{h_{1,1}}(z) = 1 + z^2\frac{2\Delta^2}{c} + 4z^2\Delta\Delta' + 2z^2(\Delta'^2 + \Delta\Delta'')c + \dots, \tag{B.6}$$

so the first term of (B.1) becomes:

$$\begin{aligned}\mathcal{F}_{h_{1,1}}(z)\mathcal{F}_{h_{1,1}}(\bar{z}) = &(z\bar{z})^2\frac{4\Delta^4}{c^2} + (z\bar{z})^2\frac{16\Delta^3\Delta'}{c} + (z^2 + \bar{z}^2)\frac{2\Delta^2}{c} \\ &+ 1 + 4(z^2 + \bar{z}^2)\Delta\Delta' + (z\bar{z})^2\left(24\Delta^2\Delta'^2 + 8\Delta^3\Delta''\right) + \dots.\end{aligned} \tag{B.7}$$

Lastly we look at the third term in (B.1). The logarithmic block at generic $c$ is given by

$$\begin{aligned}\mathcal{F}^{\log}_{h_{1,2}}(z,\bar{z}) = (z\bar{z})^{h_{1,2}(c)}\bigg(&z^2 + \bar{z}^2 + (z\bar{z}^2 + z^2\bar{z})\frac{h_{1,2}(c)}{2} + 2(z\bar{z})^2 h_{12}(c)(1 + h_{12}(c))\alpha(c) \\ &+ (z\bar{z})^2 g(c)^2\left(\lambda(c) + b_{12}(c)\ln(z\bar{z})\right) + \dots\bigg),\end{aligned} \tag{B.8}$$

where $b_{12} = b^{\text{Potts}}_{12}$ in (2.6) and we have from [4]: [14]

$$g^2(c) = \frac{\beta^4}{1024(1 - 2\beta^2)^2}. \tag{B.9}$$

---

[14] In terms of the parameters defined in [4], $\lambda(c) = \frac{2\nu}{\kappa^2 r}\left(s + \frac{\mu}{2r}\right)$ whose value at $c = 0$ is given by $\lambda = \frac{145}{18} - 10\ln 2$.

As $c \to 0$ one has

$$\mathcal{F}^{\log}_{h_{1,2}}(z,\bar{z}) = \left(z^2 + \bar{z}^2 + z^2\bar{z}^2 g^2\left(\lambda + b_{12}\ln(z\bar{z})\right)\right)\left(1 + ch'_{1,2}\ln(z\bar{z})\right) + \left(z^2\bar{z} + z\bar{z}^2\right)\frac{h'_{1,2}c}{2}$$
$$+ (z\bar{z})^2 c\left(\frac{h'_{12}}{2} + (g^2\lambda)' + (g^2 b^{\text{Potts}}_{12})'\log(z\bar{z})\right)\ldots. \tag{B.10}$$

As argued in [5], the amplitude needs to have the behavior

$$A_{aabb}(h_{1,2}) = \frac{\eta}{c} + \kappa + O(c^1), \tag{B.11}$$

which can also be seen from the necessity of canceling the simple pole $1/c$ at $(z^2 + \bar{z}^2)$ in (B.7). The contribution in the four-point function (B.1) is therefore given by:

$$A_{aabb}(h_{1,2})\mathcal{F}^{\log}_{h_{1,2}}(z,\bar{z}) = \frac{\eta}{c}\left(z^2 + \bar{z}^2 + z^2\bar{z}^2 g^2\left(\lambda + b_{12}\ln(z\bar{z})\right)\right)$$
$$+ \left(\kappa + \eta h'_{1,2}\ln(z\bar{z})\right)\left(z^2 + \bar{z}^2 + z^2\bar{z}^2 g^2\left(\lambda + b_{12}\ln(z\bar{z})\right)\right) \tag{B.12}$$
$$+ \frac{\eta}{2}\left((z^2\bar{z} + z\bar{z}^2)h'_{1,2} + (z\bar{z})^2\left(h'_{12} + (2g^2\lambda)' + \ln(z\bar{z})(2g^2 b^{\text{Potts}}_{12})'\right)\right) + \ldots.$$

Combining the expressions (B.5), (B.7) and (B.12), we see the following conditions have to be satisfied in order for the double pole and simple pole at $c = 0$ to disappear:

$$\begin{aligned}
\mathcal{O}(c^{-2}): && (z\bar{z})^2 &: r_2 + 4\Delta^4 = 0, \\
\mathcal{O}(c^{-1}): && z^2, \bar{z}^2 &: \eta + 2\Delta^2 = 0, \\
&& (z\bar{z})^2 &: 16\Delta^3\Delta' + r_1 + \eta\lambda g^2 = 0, \\
&& (z\bar{z})^2\log(z\bar{z}) &: r_2 h'_{3,1} + \eta g^2 b_{12} = 0.
\end{aligned} \tag{B.13}$$

These indeed agrees with the conditions (2.13), (2.28) and (2.30) we obtained in the main text. It is easy to check (B.13) to be true using $\Delta = h_{\frac{1}{2},0}$ and the values of $g$ and $\lambda$ at $c = 0$ (see eq. (B.9) and footnote 14).

The finite part of the $c = 0$ identity block in (B.1) is therefore given by:

$$\langle\Phi_{\frac{1}{2},0}\Phi_{\frac{1}{2},0}\Phi_{\frac{1}{2},0}\Phi_{\frac{1}{2},0}\rangle = 1 + (z^2 + \bar{z}^2)\left((4\Delta\Delta' + \kappa) + \eta h'_{12}\ln(z\bar{z})\right) + \frac{\eta h'_{12}}{2}(z^2\bar{z} + z\bar{z}^2)$$
$$+ (z\bar{z})^2\left(r_0 + 24\Delta^2\Delta'^2 + 8\Delta^3\Delta'' + \lambda g^2\kappa + \eta(\lambda g^2)' + \frac{\eta h'_{12}}{2}\right.$$
$$+ \ln(z\bar{z})\left(\frac{r_2 h''_{31}}{2} + h'_{31}r_1 + \eta\lambda g^2 h'_{12} + g^2\kappa b^{\text{Potts}}_{12} + \eta(g^2 b^{\text{Potts}}_{12})'\right) \tag{B.14}$$
$$\left.+ \ln^2(z\bar{z})\left(\frac{r_2 h'^2_{31}}{2} + \eta h'_{12}g^2 b_{12}\right) + \ldots\right) + \ldots,$$

where $\Delta = h_{\frac{1}{2},0}$ but the expression applies for generic four-point function of diagonal field $\Phi_\Delta$. This of course has to agree with the expression (3.27) which was obtained by taking the $c \to 0$ limit of the $s$-channel OPE first and then substituting the finite two-point functions at $c = 0$. To check the agreement, we now give the $f_1, f_2$ in (2.31a) whose explicit expressions were neglected there. Note that these quantities, as well as the resulting parameters $a_1, a_2$ in (2.41), are not important for the intrinsic logarithmic structure in the CFT. In particular, their values depending on the dimension of the "probing operator" $\Phi_\Delta$ and their appearance in the two-point functions (2.41) can be shifted away by a change of basis for the Jordan blocks.

However for the purpose of comparing with the four-point function explicitly, we need the following:

$$f_1 = \frac{4\Delta' h'_\Phi + \Delta h''_\Phi}{2\Delta h'_\Phi}, \quad f_2 = \frac{4\Delta^2 g' - \kappa g}{8\Delta^3 h'_\Phi}, \tag{B.15}$$

and here we take $h_\Phi = h_{31}$. The resulting parameters $a_1, a_2$ in (2.41) are given by

$$a_1 = \frac{4a_0^2}{b_{12}^2}\left(-\frac{1}{2}(b_{12}^2 h''_\Phi + b'_{12}) + \frac{b_{12}}{\Delta^2}\left(\frac{\kappa}{4} + b_{12} g' \Delta + 2\Delta\Delta'\right) - \frac{\lambda}{2}\left(h'_{12} - h'_{31}\right)\right), \tag{B.16a}$$

$$a_2 = \frac{4a_0^2}{b_{12}^2}\left(\frac{b_{12}^2}{4\Delta^4}\left(r_0 + 24\Delta^2\Delta'^2 + 8\Delta^3\Delta''\right) + \frac{\lambda\kappa}{4\Delta^2} - \frac{\lambda g g' b_{12}^2}{\Delta^2} - \frac{\lambda'}{2}\right). \tag{B.16b}$$

It is then easy to check that (B.14) indeed agrees with (3.27) by using eqs. (2.10), (2.34a) and (B.13).

## C  About self-duality

While the literature abounds in formal arguments of why diagrams for $\mathcal{V}$ should be self-dual [13, 15], it is instructive to discuss what it means in elementary terms. Restricting for simplicity to boundary theories and the Virasoro algebra, a diagram is self-dual if it is invariant under reversal of all arrows. [15] A well known example of such a diagram occurs in boundary percolation [25] with

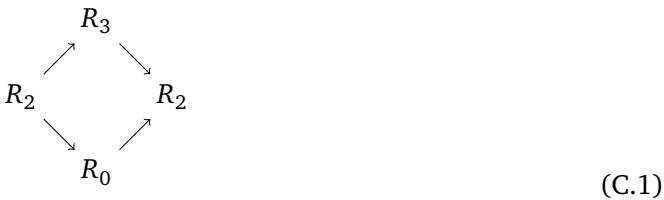

$$\tag{C.1}$$

where $R_0, R_2, R_3$ stand for Virasoro simple modules with highest weights $h_{1,1} = 0, h_{1,5} = 2,$ $h_{1,7} = 5$ in the $c = 0$ theory, and the arrows represent the action of the Virasoro algebra (so for instance, having an oriented arrow from $R_2$ to $R_3$ means it is possible to go from (some) states in $R_2$ to (some) states in $R_3$, but not the other way around). The diagram is obviously invariant under reversal of the arrows and flipping around the vertical axis.

In contrast, a diagram such as

$$R_2$$
$$\nearrow$$
$$R_0$$

$$\tag{C.2}$$

is **not self-dual**. To see what is wrong with it, let us take a state $|v\rangle$ in $R_0$, and its image $|w\rangle$ in $R_2$ under the action of some polynomial in the Virasoro generators $P(\{L_n\})$. We have

$$|w\rangle = P(\{L_n\})|v\rangle. \tag{C.3}$$

First, let us show that $|w\rangle$ has zero Virasoro-norm [16] square. This is because

$$\langle w|w\rangle \neq 0 \iff \langle P(\{L_n\})v|w\rangle \neq 0 \iff \langle v|P(\{L_{-n}\})w\rangle \neq 0, \tag{C.4}$$

---

[15]combined with horizontal or vertical flipping, depending on the convention of how one draws the diagrams

[16]This is the usual conformal norm for which $L_n^\dagger = L_{-n}$. While it is not positive definite in non-unitary theories, it is nevertheless the norm relevant to the calculation of correlation functions.

in other words, if $w$ does not have zero norm-square, it must be possible to "go back" from $w$ to $v$ (and maybe something else) by conjugate action of the Virasoro generators. But by assumption, (C.2) is all we have, therefore we reach a contradiction.

Now that we know that $|w\rangle$ has zero-norm square, we must appeal to the principle that *our theory should not have a state that is orthogonal to all other states*: otherwise the associated field would have vanishing two-point functions with all other fields in the theory, and therefore be redundant and could be factored out. This means therefore that there must exist another state $|w'\rangle$ such that $\langle w|w'\rangle \neq 0$. Note that, by general CFT principles, $|w'\rangle$ must have the same conformal weight as $|w\rangle$. So now we have

$$\langle w|w'\rangle \neq 0 \iff \langle P(\{L_n\})v|w'\rangle \neq 0 \iff \langle v|P(\{L_{-n}\})w'\rangle \neq 0, \tag{C.5}$$

so there must be another arrow going from $|w'\rangle$ onto $|v\rangle$.

$$
\begin{array}{ccc}
|w'\rangle & & |w\rangle \\
& \searrow \quad \nearrow & \\
& |v\rangle &
\end{array}
\tag{C.6}
$$

so once again we reach a contradiction if we suppose (C.2) is all we have.

Using this kind of argument, it is easy to see that, whenever we have a diagram and a pair of modules $R, R'$ with an arrow going from $R$ to $R'$, we must have another copy of $R'$ - denote it by $R''$ - with an arrow going from that copy to $R$. The subset $R, R', R''$ is then invariant by duality. The argument generalizes to more complicated cases.

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
