# Peer review of "A note on the identity module in $c=0$ CFTs"

_SciPost Physics, doi:SciPost Phys. 12, 100 (2022)_

## Round 1 · Referee Report · Anonymous (Referee 2) · 2021-12-20

Strengths

  1. Detailed and precise work
  2. Clear presentation

Weaknesses

  1. Given the technicality of the paper, the target audience is pretty small

Report

The paper studies in detail the structure under Virasoro of fields appearing in the identity blocks for some logarithmic theories with vanishing central charge. The precise structure is worked out using the c->0 limit, which is unique and well defined in the theories considered.

The paper is technical but clear. I recommend publication in SciPost, after few very minor changes.

Requested changes

  1. Eq (2.5): how are the normalizations fixed? For $b_{12}$ in eq (2.6) to have meaning that should be specified. I assume it has to do with the two point function of $X$ but I wasn't able to find this.

  2. Is there an argument why no operators would have factors such as $1/c$? Eq (2.18) assumes that, for any n-point function, terms such as $c t$ always vanish, meaning that other operators cannot involve inverse factors of $c$ in general. An explanation would be great.

  3. Why does (2.21) follow from (2.20)? It could be that $\bar \partial X $ has vanishing 2pf but not higher npf, given that the theory is not unitary. If there is some other reason for (2.21) it would be good to have it written down.

---

## Round 1 · Referee Report · Anonymous (Referee 1) · 2021-12-20

Report

This paper addresses the structure of the indecomposable Virasoro-times-Virasoro module containing the identity field in the CFTs believed to describe the appropriate limits of the $O(n)$ and Potts models. This is an old problem that has not received as much attention as it should. The present paper argues convincingly that the states in this module of conformal weights $(h,\bar{h})$, with $h,\bar{h} \le 2$, include Jordan blocks for the Virasoro mode $L_0$ of ranks $2$ and $3$. The deeper structure is not investigated.

The methodology is a straightforward, though technically involved, generalisation of the well known $c \to 0$ catastrophe argument. This combines some recent bootstrapping results with basic physical principles. The results are shown to be self-consistent and extend a recent proposal of Nivesvivat and Ribault. (They also extend several other earlier proposals, though this is not discussed in detail.)

There is a lot to learn from this paper, including technical asides that one may not have thought of prior to commencing calculation. The results are interesting and warrant publication and I recommend it warmly with some minor rewrites. I also hope that this paper stimulates further investigation of further (non-chiral) logarithmic structures.

Requested changes

I have a short list of comments and suggestions: 0. The paper is generally well written, but there are local occurrences where this is not the case. I suggest a thorough proof reading to catch certain paragraphs in which the grammar obscures the arguments being made. 0'. It may not be my place to say, but the usage of Jordan cell'' instead of the much more standardJordan block'' throughout is known to be a mistranslation on the part of Gurarie that has propagated through the community. I suggest changing it. 1. The abstract seems to claim that non-trivial CFTs with $c=0$ are logarithmic. I wonder in what sense the product $M(4,5) \otimes M(5,8) \otimes M(5,8)$ of Virasoro minimal models is trivial. 2. There is occasional mention of Loewy diagrams'', eg in the first paragraph of Sec 2.1. I think it is worth mentioning that a Loewy diagram does not fix the action of the algebra completely (as suggested in that paragraph) and also that it is not the same as the pictures we meet in (3.11) and Figure 2. In fact, I cannot find any Loewy diagrams in the paper, though I think adding one to summarise the paper's main result would be very welcome! 3. In the introduction and Sec 2.1, fields $t$ and $\bar{t}$ appear without much information. They end up being defined in Sec 2.2. However, non-experts are likely to be very confused until they read the later section --- the notation even suggests that $t$ is holomorphic and $\bar{t}$ is antiholomorphic, neither of which is true. I suggest that more information about these fields is included in Sec 2.1 to avoid such potential confusion. 4. Before (2.6), a parenthetical reminds us that certain constants appearing in correlators depend on the choice of logarithmic partner field. However, this is only the case if the logarithmic coupling is non-zero. [This is the case for almost all $c$ here, but we don't actually know that at this point.] 5. (2.6) itself consists of some non-trivial statements about chiral logarithmic couplings. I'd like to see a derivation or a citation here, please. 6. Footnote 3 is curious. It appears to say that linear combinations of fields of different dimensions are problematic (which is of course untrue). Maybe this could be clarified. 7. (2.13) is noted to be an assumption required for the analysis, at least for the $O(n)$ models. I think it would be appropriate to explicitly note this assumption, along with others that come later, when describing the results in the introduction and conclusions. 8. One thing I found confusing was the authors' use of $=$ to mean not only$=$'' but also $=$ once a $c\to0$ limit is taken''. The reader is warned about this explicitly, but I think it would significantly improve the exposition if$\overset{c\to0}{=}$'' was used for the latter (as it is in (2.20)). Also, one comes across big-$\mathcal{O}$ notation, eg (2.29). Maybe just one precise way, $c\to0$ or $\mathcal{O}$, of explaining such equations would suffice? 9. Sec 2.3 is also, in my opinion, unnecessarily complicated by not explaining the appearance of new symbols until well after they appear. Here, I refer to (2.27) in which $\Phi_2$ has been previous introduced, but $\Phi_1$ and $\Phi_0$ are not until (2.32). This destroys the reader's flow as they search the previous text in vain for the latter's definitions. 10. (2.29) and (2.31) are, like (2.13), assumptions that should be mentioned in both the introduction and conclusions when asserting the fine results of this paper. 11. The way it is written, there seems to be no reason to give (2.32). I suggest rewriting this part of the text and adding a forward reference to the later results that depend on these expressions. 12. (2.42) is one of the key results it seems, the rank $3$ Jordan block logarithmic coupling. But it is stated without derivation. Possibly (2.44) is actually the derivation, but I could not be sure if that was the intention. I also note that the "easy to verify" claim leading to (2.43) doesn't appear to be possible to verify without such a derivation. Please rewrite these paragraphs. 13. In a few places, the authors refer to a Hilbert space. It isn't clear which Hilbert space they are referring to, but perhaps it should be mentioned that any space of states including non-trivial Jordan blocks for $L_0$ cannot form a Hilbert space as the $L_0$-eigenvector has zero norm. 14. The first two paragraphs of Sec 3.1 were a little difficult to understand and I suggest a rewrite. In particular, did the authors forget the Jordan block for the barred fields? 15. After (3.9), it may be wise to spell out what is meant by change of basis'' and what it implies for other calculations. 16. (3.11) suddenly uses kets, without comment, whereas all previous equations do not. Actually, kets are introduced in the following section, Eq. (3.19), so perhaps this is a typo? 17. As (3.11) is a first approximation to one of the final results, it may be useful to indicate precisely what is being pictured here. A casual glance leads one to question what it means if an arrow is missing --- eg there is no arrow representing the action of $A$ on $\Psi_2$: does that mean that the result is $0$? I think not, because there are many $L_0$ arrows missing which definitely don't give $0$. 18. After (3.11), the phraseindependent of the normalization'' is used, referring to $a_0$. Could the authors explain what they mean? I would assume that normalization'' refers to some measure of the size of some field, eg $\Psi_2$ in (2.41a). However, multiplying $\Psi_2$ by $7$ clearly changes $a_0$, so the authors must mean something different. 19. Before (3.22), the authors mention the logarithmic conformal block of the identity module. Could they please explain what they mean, precisely? I suspect that they mean the right-hand side of (3.22), which is unfortunately not equal to the 4-pt function on the left-hand side but rather a projection of it. 20. (3.33) is curious: why would one expect both calculations to give $\Psi_1$ on the nose? Perhaps one thinks to tune $\alpha$ to make it so, but then why not different $\alpha$ for the two calculations? It wasn't clear to me if the authors weren't making an implicit assumption here. Could they clarify? 21. In (3.39) and below, there seems to be a systematic typo in which $|A\bar{t}>$ is written instead of $A|\bar{t}>$, etc. 22. Figure 2 isthe final structure'' of the identity module, up to fields with $h,\bar{h} \le 2$. However, it is missing many arrows describing the action of $L_0$. The state $\tilde{\Psi}_1$ in the middle has no arrows coming in and out, so the final structure appears to be indicating that it is a primary field that doesn't belong to the non-chiral vacuum module, but another direct summand. Please add some discussion to clarify what is being asserted here. [Some of this is clarified in Sec 3.3.1, but this is another instance in which the fact that there is clarification to come needs to be stated clearly at the right place. Otherwise, it cannot really be ``the final structure''.] 25. (3.47) is preceded by a reference to Sec 2.3, which should really be a reference to an equation number. 26. Sec 4 suggests that the structure of the non-chiral vacuum module has been fixed for fields with $h+\bar{h}\le4$, which is stronger than what was asserted in previous sections.

---

## Round 2 · Referee Report · Anonymous (Referee 2) · 2022-2-11

Strengths

  1. Detailed and precise work
  2. Clear presentation

Weaknesses

  1. Given the technicality of the paper, the target audience is pretty small

Report

I am happy with the answer and changes the author did for point 1) and 3), but I'm not satisfied with the answer for point 2).

Clearly an object such as $\frac{1}{c} \phi$ by itself is not well defined in the $c \to 0$ limit. However, it could be that there's some operator $\mathcal{O} =\# \psi + \frac{1}{c}\phi$ which is well defined in the $c \to 0$ limit for some appropriate prefactor of some other operator $\psi$. This is what happens for $t$, see eq (2.9); from my understanding it could happen for other operators as well. Now, if I consider a correlation function of the type $\langle t \mathcal{O} \ldots \rangle $, what tells me that I will not have cancellations between powers of $c$ so that it will remain finite as $c \to 0$?

Requested changes

Better explanation of point 2)

---

## Round 2 · Referee Report · Anonymous (Referee 1) · 2022-2-14

Report

I thank the authors for their updates/comments and recommend this version for publication.

---

## Round 2 · Author Response

Dear Editor,

We would like to thank you for considering the publication and thank the referees for reading the manuscript so carefully, posing many interesting questions/comments and making valuable suggestions for improvement of the paper. Please see below for the replies to the referees and changes made under the referees' suggestions.

Best regards,
Yifei He and Hubert Saleur

---

## Round 2 · List of Changes

Warnings issued while processing user-supplied markup:

  • Inconsistency: plain/Markdown and reStructuredText syntaxes are mixed. Markdown will be used.
    Add "#coerce:reST" or "#coerce:plain" as the first line of your text to force reStructuredText or no markup.
    You may also contact the helpdesk if the formatting is incorrect and you are unable to edit your text.

Anonymous Report 2 on 2021-12-20

1, Indeed, as the referee pointed out, the normalization of $\Psi$ is fixed by the normalization of $X$, namely that its two-point function is normalized to 1. This can be seen straightforwardly by considering the relation $A^{\dagger}\Psi=b_{12}\bar{X}$ and write: \begin{equation} \langle A\bar{X}|\Psi\rangle=\langle\bar{X}|A^{\dagger}\Psi\rangle=b_{12}\langle\bar{X}|\bar{X}\rangle=b_{12} \end{equation} and use the relation between the inner product of the states with the CFT two-point function. Regarding this point, we have added footnote 2 with comments.

2, The standard argument that no operators can have factors such as $\frac{1}{c}$ is as follows: Suppose there is an operator which takes the form $\frac{1}{c}\phi$ where $\phi$ has order 1 two-point functions, then the field is not well-defined at $c=0$ as its two-point function diverges, and therefore it does not make sense to consider operator insertion of $\frac{1}{c}\phi$ into correlation functions. We have added a brief comment on this in footnote 7.

3, A way to see that a field is chiral is to look at its two-point function and in particular its holomorphicity. This different from directly looking at the anti-chiral conformal dimension, which might vanish without excluding some logarithmic dependency. An example of this is the logarithmic field $t(z,\bar{z})$: While its anti-chiral conformal dimension vanishes, it is a non-chiral field and this can be seen from the two-point function of $\bar{\partial}t$ as we have written in eq. (2.22). It is by this standard that we claim $-\sqrt{-\frac{c}{2}}\bar{\partial}X$ a chiral field: from (2.20), the two-point function of this field goes as \begin{equation} \langle\Big(-\sqrt{-\frac{c}{2}}\bar{\partial}X\Big)(z,\bar{z})\Big(-\sqrt{-\frac{c}{2}}\bar{\partial}X\Big)(0,0))\rangle=\mathcal{O}(c^2) \end{equation} vanishing at $c=0$. This agrees with our identification with $T(z)$ which is further expected from lattice considerations. It would be however interesting to study further to find a more rigorous argument (including also the physical interpretation of this identification) which we leave for future work.

===================================================================

Anonymous Report 1 on 2021-12-20

0', Following the referee's comments, we have changed all the names "Jordan cells" to "Jordan blocks". Before Gurarie, we note that some old french books used "cellule de Jordan" quite commonly. 1, We are not considering the case of $c=0$ CFTs that are constructed by tensor products. Indeed, one can resolve the $c\to 0$ catastrophe in tensor product CFTs, for example by taking two non-interacting CFTs with equal and opposite central charges, and introducing a chiral field $t(z)$. However in this case the algebra of $T$ and $t$ form are trivial. A nice argument can be found in section II A of reference [20] which we refrain from repeat here. 2, To avoid confusing the readers between the diagrams we draw in the paper and Loewy diagrams, we have suppressed mentioning Loewy diagrams" throughout the paper. 3, To avoid the potential confusion about $t,\bar{t}$ as the referee mentioned, we have added the two-point functions of $(t,T)$ in the introduction. (page 3) 4, The constant in the two-point function of the logarithmic field (for example $\Psi$ in eq. (2.5a)) depends on the choice of basis in the Jordan block. As we have commented in the manuscript, this constant can be modified by a change of basis. Of course, as the referee pointed out, this is only the case for non-zero logarithmic coupling. (With zero logarithmic coupling, there is no logarithmic operator and thus no Jordan block.) 5, Eq. (2.6) are the logarithmic couplings of the rank-2 Jordan block (figure 1) at generic $c$ as recently obtained in references [1,4]. We would like to point out that they are non-chiral logarithmic couplings, contrary to what the referee said. By the request of the referee, we have added footnote 3 for a clear reference to both references [1,4] where the derivations can be found. 6, The linear combination of fields with different dimensions do not correspond to scaling fields (for example, their correlation functions do not have a scaling behavior with certain exponent). It is in this sense that we are referring to this asdimensionally problematic'' in footnote 5. 7, The singularity cancellation conditions (2.13), (2.28) and (2.30) are obtained by requiring the finite behavior of the $c=0$ theory which is a key to our analysis. (This dates back to Gurarie's original analysis on the $t$ field although we have studied beyond that.) To make this clear, as the referee suggested, we have added the following sentence to the introduction (second to last paragraph on page 2): “By requiring the finiteness of two-point functions at $c=0$, we obtain singularity cancellation conditions (2.13), (2.28) and (2.30), which allow us to establish the existence of a rank-three Jordan block of fields of weight $h=\bar{h}=2$ when $c=0$ with the bottom field being $T\bar{T}$, and determine the corresponding universal logarithmic coupling.” 8, We have followed the referee's suggestion to use the notation $\overset{c\to 0}{=}$ at various places when we mean in the $c\to 0$" limit (in addition to the comments we had made before). This appears in eqs. (2.17), (2.18), (2.19), (2.21), (2.37) and (2.38). In eq. (2.20) we have kepted the big-$\mathcal{O}$ notation but removed $\overset{c\to 0}{=}$ as the referee suggested. 9, We have rewritten the beginning of section 2.3 to make the flow of the reading more smooth. In this part, we would like to first focus on the field $\Phi_2$ and obtain the conditions (2.28) and (2.30) as required by the finiteness of its 2-point function. This is why we postpone the definition of $\Phi_1,\Phi_2$ to avoid interruption. Now we have postponed writing down the logarithmic OPE (2.32) after the definition of the fields $\Phi_1,\Phi_0$ to avoid the confusion as the referee mentioned. We believe this is a more logical order for writing this part. 10, This question is related to question 7 and we have answered both above. 11, This is related to the answer 9 above. We believe it is good to keep the definition (2.31) (previously (2.32)) in order for the readers to check straightforwardly the logarithmic OPE (2.32). Therefore we have kept these definitions. 12, Based on the referee’s suggestion, we have slightly rewritten the part between eq. (2.40) and (2.44) to make it more clear. The expression of $a_0$ is by direct calculation using the definitions of the fields $\Psi_{0,1,2}$. Using the current expression of (2.42), it should be straightforward to verify (2.43) which makes it clear that the logarithmic coupling $a_0$ is identical for Potts and $O(n)$ models. Eq. (2.44) is not a derivation, but rather an interesting observation which makes it more manifest of the claim above. It remains an interesting open question whether there is a structural derivation of (2.44). 13, By Hilbert space, we meant the CFT state space. Indeed as the referee pointed out, there are zero norm states, and therefore we have changed all the places we usedHilbert space" to CFT state space" to avoid any confusion. 14, We have rewritten the beginning of section 3.1 to make it clear the procedure of applying conformal invariance to obtain the actions of $L_n,\bar{L}_n$. On the other hand, the Jordan block of the barred fields $(\bar{t},\bar{T})$ is the same as the ones for $(t,T)$ with the replacement $L_0\to \bar{L}_0$. We have added a brief comment on this in footnote 9. 15, As the referee suggested, we have added comment on this in footnote 10. 16, Before depicting (3.11), we have mentioned in the last paragraph on page 11 ofusing state-operator correspondence” and then switched to the ket notation. This allowed us to directly depict the structures on the CFT states under Virasoro algebra based on the actions we obtained earlier in this section on the operators. 17, The structure depicted in (3.11) is a result of imposing simply conformal invariance on the logarithmic OPE (2.45) for all fields up to $h,\bar{h}=2$. To make this clear, we have added comments after the figure as the referee suggested. 18, As the referee mentioned, one can consider the normalization of the field eg. $\Psi_2$ as measured by its 2-point function eq. (2.41a), and indeed, multiplying $\Psi_2$ by a constant would change the constant appearing in the 2-point function. In this sense, the constants appearing in the 2-point function are not normalization-independent quantities. However, for the rank-3 Jordan cell, if one modifies the normalization of $\Psi_2$, one has to do the same for $\Psi_1$ and $\Psi_0$ (with the same normalization factor) in order to preserve the form of their two-point functions as in eq. (2.41) which is required by conformal invariance. In this case. the $a_0$ appeared in the algebraic relation (3.13) remains unchanged, and it is in this sense that the $a_0$ here is normalization independent. (We have chosen a normalization for the fields $\Psi_{0,1,2}$ such that the $a_0$ appearing in (3.13) agrees with the constant in (2.41) but the constants in (2.41) can be modified by a different normalization while the one in (3.13) cannot.) We have added footnote 11 to comment on this. 19, Indeed, as the referee pointed out, the rhs of eq. (3.22) needs to be reinterpreted. By logarithmic conformal block of the identity module, we mean the four-point function projected onto the subspace of the CFT state space made of the states constructed from the identity module by Virasoro algebra. We have rewritten the part between eq. (3.22) and (3.24) to make this clear. In the meantime, we interpret the rhs of (3.22) as summing over all the states in the CFT state space which in particular includes the identity module. 20, The logic of (3.34) (originally (3.33)) is as following: from the computation of Gram matrix in (3.21), we see that $\Psi_1$ has non-zero inner product with $\Psi_1,\Psi_2$ which leads to (3.32) from (3.31). In (3.33), we then see that the option of $\Psi_2$ violate self-duality and therefore choose $\Psi_1$. The normalization in (3.31) and comparison with (3.21) allows to conclude that this is precisely $\Psi_1$ (no rescaling by some constant) In this derivation, one does not tune $\alpha$, just leave it as an unknown number which is fixed only later in (3.36). 21, We have corrected the typos of $A|\bar{t}\rangle$ as suggested by the referee. 22, Indeed as the referee pointed out, we have refrained from drawing the actions of $L_0,\bar{L}_0$ in figure 2. These actions are explicitly stated in eqs. (3.4) and (3.5) and one reason to not draw them is to avoid making the figure 2 too messy-looking. We have added comments after the figure, also commenting and referencing to the next subsection on more details of the state/operator $\tilde{\Psi}_1$. We hope this is enough to make things clear for the readers. 25, We have added the reference to the equations that defines the field $\Psi_1$ as suggested by the referee. 26, We have modified the phrase What happens for $h+\bar{h}>4$ remains to be explored" toWhat happens for $h,\bar{h}>2$ remains to be explored".

---

## Round 3 · Referee Report · Anonymous (Referee 2) · 2022-2-15

Report

I am now happy with the explanation of point 2) and recommend publication

---

## Round 3 · Author Response

Dear Editor,

Thank you for the consideration of the publication and the referee for further questions.
Please see below for our answer. We hope this addresses the referee's concern.

Best regards,
Yifei He and Hubert Saleur

---

## Round 3 · List of Changes

One of the basic assumption is that for an operator to be well defined in the $c=0$ theory, it must have finite correlation functions. This is the key to the $c\to 0$ analysis. Indeed, one can consider the case as the referee mentioned of an operator defined using the generic $c$ fields $\psi,\phi$ as $\mathcal{O}=#\psi+\frac{1}{c}\phi$, but for it to be a well-defined $c=0$ operator it must have finite correlations at $c=0$. One can analyze its 2-point function in a similar way as $t$. (in this case it will result in a logarithmic operator with certain conditions on $#$) Perhaps the real concern is whether the two-point function analysis guarantees the finiteness of all other correlation functions of $O$ (or $t$) defined this way. To make this generic claim requires further studies. However in the special case of the operator $t$, we know for example that its 3-point function is also well-defined as studied in reference [18].
To summarize, in writing eq. (2.18), we are assuming that $\ldots$ involves only well-defined operators at $c=0$ in the above sense and therefore $\langle t(z,\bar{z})\ldots \rangle$ is finite by this assumption. We have slightly modified the footnote 7 to make this clear.

---

## Editorial Decision

published